# Integrating and formatting biomedical data as pre-calculated knowledge graph embeddings in the Bioteque

Adrià Fernández-Torras [1], Miquel Duran-Frigola [1,2], Martino Bertoni [1], Martina Locatelli [1] & Patrick Aloy [1,3] ✉

Biomedical data is accumulating at a fast pace and integrating it into a unified framework is a major challenge, so that multiple views of a given biological event can be considered simultaneously. Here we present the Bioteque, a resource of unprecedented size and scope that contains pre-calculated bio-medical descriptors derived from a gigantic knowledge graph, displaying more than 450 thousand biological entities and 30 million relationships between them. The Bioteque integrates, harmonizes, and formats data collected from over 150 data sources, including 12 biological entities (e.g., genes, diseases, drugs) linked by 67 types of associations (e.g., 'drug treats disease', 'gene interacts with gene'). We show how Bioteque descriptors facilitate the assessment of high-throughput protein-protein interactome data, the prediction of drug response and new repurposing opportunities, and demonstrate that they can be used off-the-shelf in downstream machine learning tasks without loss of performance with respect to using original data. The Bioteque thus offers a thoroughly processed, tractable, and highly optimized assembly of the biomedical knowledge available in the public domain.

Systematic measurements of biological samples through omics technologies, together with efforts to distil the scientific literature into structured databases, are providing an ever-growing corpus of biomedical and biomolecular information[1]. Indeed, the data stored in the EMBL-EBI has increased sixfold in the last few years, from 40 petabytes in 2014 to over 250 in 2021[2]. Associated with this phenomenon, a variety of nomenclatures have been proposed, along with identifiers, levels of resolution (e.g., protein isoforms or gene splice variants) and experimental conditions, making data integration and harmonization across platforms a challenging step[3]. As a result, even though as many as 1641 resources were listed in the 2021 Online Molecular Biology Database Collection[4], only a small portion are broadly used, and hundreds remain isolated with their own particular formats[5,6]. Aware of the situation, several initiatives have emerged to standardize biological data by establishing common vocabularies and formats. For instance, the pioneering Harmonizome[7] was able to

integrate knowledge from several gene-centric databases by representing data (e.g., gene expression, disease genetics, etc.) in a simple discretized format that was applicable to each type of data.

Nowadays, in an attempt to capture the complexity of biological systems, multiple omics profiles are often measured simultaneously (i.e., trans-omics analyses)[8,9] so that complementary views of a given phenotype or event can be considered in parallel and as a whole[10]. However, current methods mainly adapt and combine existing strategies developed to analyse individual omics data, and often the net result is that most conclusions are drawn from the most informative single data type, while the rest are used as support. It is thus fundamental to devise strategies able to capture the coordinated interplay of the many regulatory layers present in biological systems. Himmelstein et al. suggested the use of knowledge graphs (KG) as a tool to integrate heterogeneous biomolecular data[11,12]. In a biomedical KG, nodes represent biological or chemical entities (e.g., genes, cell lines, diseases, drugs,

[1]Institute for Research in Biomedicine (IRB Barcelona), The Barcelona Institute of Science and Technology, Barcelona, Catalonia, Spain. [2]Ersilia Open Source Initiative, Cambridge, UK. [3]Institució Catalana de Recerca i Estudis Avançats (ICREA), Barcelona, Catalonia, Spain. ✉e-mail: patrick.aloy@irbbarcelona.org

etc.), and edges capture the interactions or relationships between them (e.g., 'drug treats disease' or 'cell upregulates gene'). This concept has recently been expanded to include clinical entities[13].

However, large biomedical networks are intractable by conventional graph analytics techniques[14], thus prompting the development of dimensionality reduction techniques that learn numerical feature representations of nodes and links in a low dimensional space (aka network embeddings). As a result, network embeddings reduce the dimensionality of the data while preserving the topological information and the connectivity of the original network[15]. Moreover, the vectorial format of the nodes resulting from network embedding approaches is better suited as an input for machine learning algorithms. For instance, Zitnik and Leskovek presented a set of protein embeddings that consider the protein interactions within each human tissue, as well as inter-tissue relationships, and showed their potential to predict tissue-specific protein functions[16]. Later on, the same authors embedded several networks (i.e., protein–protein, drug–target and disease–gene interactions) to explore the mechanisms of drug action[17]. Recently, Cantini et al. evaluated the capacity of several dimensionality reduction methods to integrate continuous multi-omics data (e.g., gene expression, copy number variation, miRNAs and methylation)[18], assessing their ability to preserve the structure of the original data and their prediction performance in different tasks. Overall, embedding-based descriptors provide a

scalable and standard means to capture complex relationships between biological entities and they integrate the myriad of omics experiments associated with them[19,20].

To make biomedical knowledge embeddings available to the broad scientific community, we have developed the Bioteque, a resource of unprecedented size and scope that contains pre-calculated embeddings derived from a gigantic heterogeneous network (more than 450k nodes and 30M edges). The Bioteque harmonizes data extracted from over 150 data sources, including 12 distinct biological entities (e.g., genes, diseases, compounds) linked through 67 types of relationships (e.g., 'compound treats disease', 'gene interacts with gene'). We demonstrate that Bioteque embeddings retain the information contained in the large biological network and illustrate with examples how this concise representation of the data can be used to evaluate, characterize and predict a wide set of experimental observations. Finally, we offer an online resource to facilitate access and exploration of the pre-calculated embeddings (https://bioteque.irbbarcelona.org).

## Results

### A comprehensive biomedical knowledge graph (KG)

To build a KG that integrates biological and biomedical knowledge available in the public domain, we first defined the basic entities (nodes) of the network and the relationships between them (edges). As shown in Fig. 1a, the resource is gene-centric.

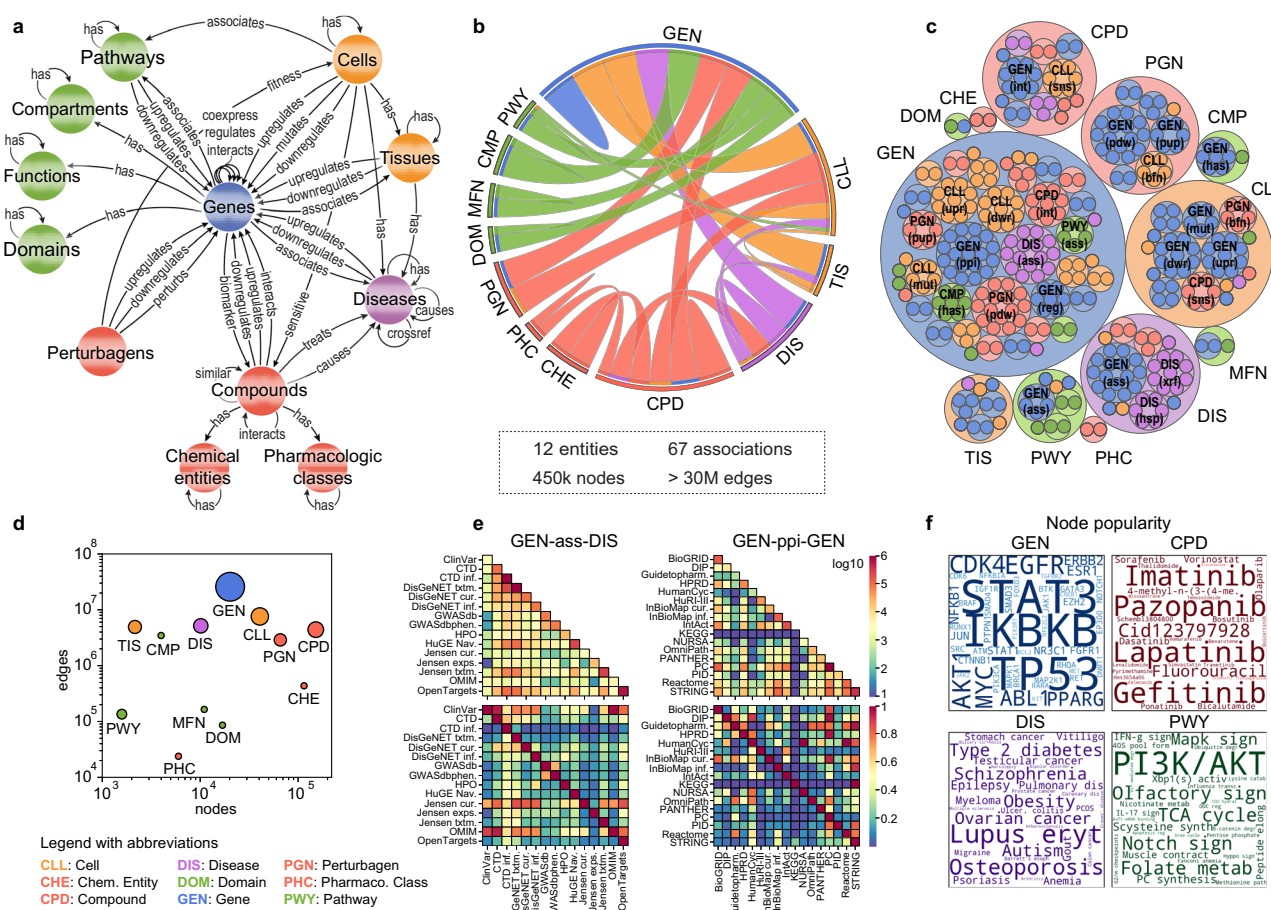

**Fig. 1 | Building the Bioteque knowledge graph (KG). a** Metagraph of the Bioteque, showing all the entities and the most representative associations (metaedges) between them. **b** Circos plot representation of the KG, showing the relationships between nodes. **c** Treeplot showing the number of datasets used to construct each metaedge. **d** Total number of nodes (x-axis) and edges (y-axis) available for each entity type. The size of the circles is proportional to the number of metaedges in which the entities participate. **e** Number of edges (top row) and

overlap (bottom row) between the datasets inside the 'gene associates with disease' (GEN-ass-DIS, left) and 'protein interacts protein' (GEN-ppi-GEN, right) associations. **f** Most popular nodes in the KG within the gene (GEN, blue), compound (CPD, red), disease (DIS, purple) and pathway (PWY, green) universe. Dataset associations were de-propagated across the corresponding ontologies (when possible) before computing the popularity of the nodes. A propagated version of this plot is shown in Supplementary Fig. 1.

**Table 1 | Biological and chemical entities in the knowledge graph (KG)**

| Metanode | Abbreviation | Nodes | Metaedges | Edges | Example 1 | Example 2 |
|---|---|---|---|---|---|---|
| Cell | CLL | 40,681 | 15 | 7,512,366 | CLL-upr-GEN | CLL-mut-GEN |
| Cellular component | CMP | 3992 | 2 | 3,461,731 | GEN-has-CMP | CPD-hsp-CMP |
| Chemical entity | CHE | 115,002 | 2 | 435,011 | CHE-hsp-CHE | CHE-hsp-CPD |
| Compound | CPD | 153,279 | 12 | 5,713,785 | CPD-int-GEN | CPD-trt-DIS |
| Disease | DIS | 10,144 | 10 | 5,037,293 | GEN-ass-DIS | CPD-cau-DIS |
| Domain | DOM | 16,913 | 2 | 85,747 | GEN-has-DOM | DOM-hsp-DOM |
| Gene | GEN | 20,229 | 42 | 25,788,255 | GEN-ppi-GEN | GEN-pho-GEN |
| Molecular function | MFN | 11,006 | 2 | 164,447 | GEN-has-MFN | MFN-hsp-MFN |
| Pathway | PWY | 1585 | 4 | 133,851 | GEN-ass-PWY | PWY-hsp-PWY |
| Perturbagen | PGN | 66,988 | 7 | 2,889,047 | PGN-bfn-CLL | PGN-gfn-CLL |
| Pharmacological class | PHC | 6072 | 2 | 31,004 | CPD-has-PHC | PHC-hsp-PHC |
| Tissue | TIS | 2157 | 8 | 4,928,112 | GEN-ass-TIS | TIS-upr-GEN |

We show the number of nodes, metaedges and edges contained in the KG for each metanode, as well as some examples of metaedges.

Thus, genes and gene products (GEN) are represented in the centre of the KG scheme and are involved in most associations. To better characterize genes and proteins, we collected their molecular function (MFN), cellular component localization (CMP), functional structure or domains (DOM), and biological processes or pathways (PWY). Additionally, we included information on cell lines (CLL), one of the most studied entities in biology, as well as their anatomical ensembles, namely the tissues (TIS). Analogously, chemical compounds (CPD) are depicted together with pharmacological classes (PHC) and chemical entities (CHE), two common vocabularies for medicinal compounds. Diseases (DIS) are abnormal conditions that have been widely studied in various fields, giving rise to a wide diversity of interactions between different nodes. Furthermore, although CPD and DIS are two of the major perturbational agents found in repositories like GEO[21] and LINCS[22], we also considered other biological entities such as miRNA, shRNA and overexpression vectors that can also act as perturbagens (PGN). To connect the entities in the Bioteque, we defined 67 types of associations reflecting biological relationships between them. An example of such an association would be a gene that is associated with a given pathway (GEN-ass-PWY) and might be downregulated in a certain cell (GEN-dwr-CLL) or tissue type (GEN-dwr-TIS), or a drug compound that is used to treat a disease (CPD-trt-DIS). A comprehensive list of all the biological and chemical entities included in the Bioteque, as well as the different associations, are summarized in Fig. 1a and Table 1 and provided in Supplementary Data 1 and 2.

Having defined the biological entities and their interactions, we populated the Bioteque with data collected from representative datasets and resources. We first incorporated data from the Harmonizome[7], the most complete compendium of biological datasets to date, and added data from another 100 reference datasets. Each dataset was mapped to the KG scheme (or metagraph) depicted in Fig. 1a. Inspired by the Harmonizome strategy, we processed each dataset separately following author guidelines, when possible ("Methods"). In brief, we binarized continuous data so that it could be represented in a network format, and we standardized identifiers from multiple sources.

The current version of the KG contains over 450k nodes, belonging to 12 types of biological entities (metanodes), and over 30M edges, representing 67 types of relationships (metaedges) (Fig. 1b). In general, the size of our KG is comparable to other recently published biomedical KGs[13,23–25]. In fact, taking as a reference the comparison made by Bonnet et al.[26], our KG is the most comprehensive in the number of processed datasets, the second most comprehensive with respect to entities, edges, and relation types, and the third regarding entity types (Supplementary Table 1). Not surprisingly, genes and

proteins account for most of the edges (25M) and metaedges (42) in the graph (Fig. 1c, d). In terms of the number of reference datasets, protein interactions (GEN-ppi-GEN) and gene-disease associations (GEN-ass-DIS) are the most represented metaedges, supported by 17 and 15 datasets, respectively (Fig. 1c). A comparison of data extracted from each dataset revealed that, although there is some overlap, most sets cover distinct associations, probably due to differences in the focus of the underlying experiments (i.e., physical[27] vs. functional[28] PPIs or drug-driven[29] vs. genomics-driven[30] gene associations) (Fig. 1e).

**Calculation of network embeddings across the KG**

To integrate the biological knowledge gathered, we devised an approach to obtain, for a given node in the KG, a set of embeddings capturing different contexts defined by one or more types of relationships between this node and other entities (Fig. 2a). For example, the pharmacological context of a certain compound can be captured by 'compound interacts with protein' associations, while its clinical context may be captured by 'compound treats disease' links. The embedding procedure is as follows. We first define the types of biological entities (metanodes) to be connected and the sequence of relationships (metaedges) between them that we wish to explore. This sequence of relationships is called metapath. We then systematically examined all possible paths from the source and target nodes of the metapath, downweighting highly connected nodes to ensure exhaustive exploration of the network[11]. This step yields a simplified homogeneous (when source and target metanodes belong to the same type) or bipartite (when source and target metanodes belong to different types) graph that can be explored with conventional network embedding techniques. We chose to use a random walk method, where the trajectories of an agent that explores the network are retained and eventually fed into a text-embedding algorithm[31]. As a result, for each node in the network, a 128-dimensional vector (i.e., an embedding) is obtained, so that similar vectors are given to nodes that are proximal in the network. During this process, we mostly keep different datasets separately (i.e., without merging equivalent networks in different sources) to preserve the original information captured in them[32]. A more detailed description of the protocol is provided in the "Methods" section.

We have created a resource of pre-calculated biomedical embeddings, the Bioteque, where we have exhaustively considered most metapaths of length 1 and 2 extracted from the KG (i.e., direct connections between source and target nodes, or with one intermediate node between them). In addition, we have curated a collection of 135 metapaths of length ≥3. Overall, the Bioteque currently holds a total of 81, 785, and 175 embeddings of length 1, 2, and ≥3,

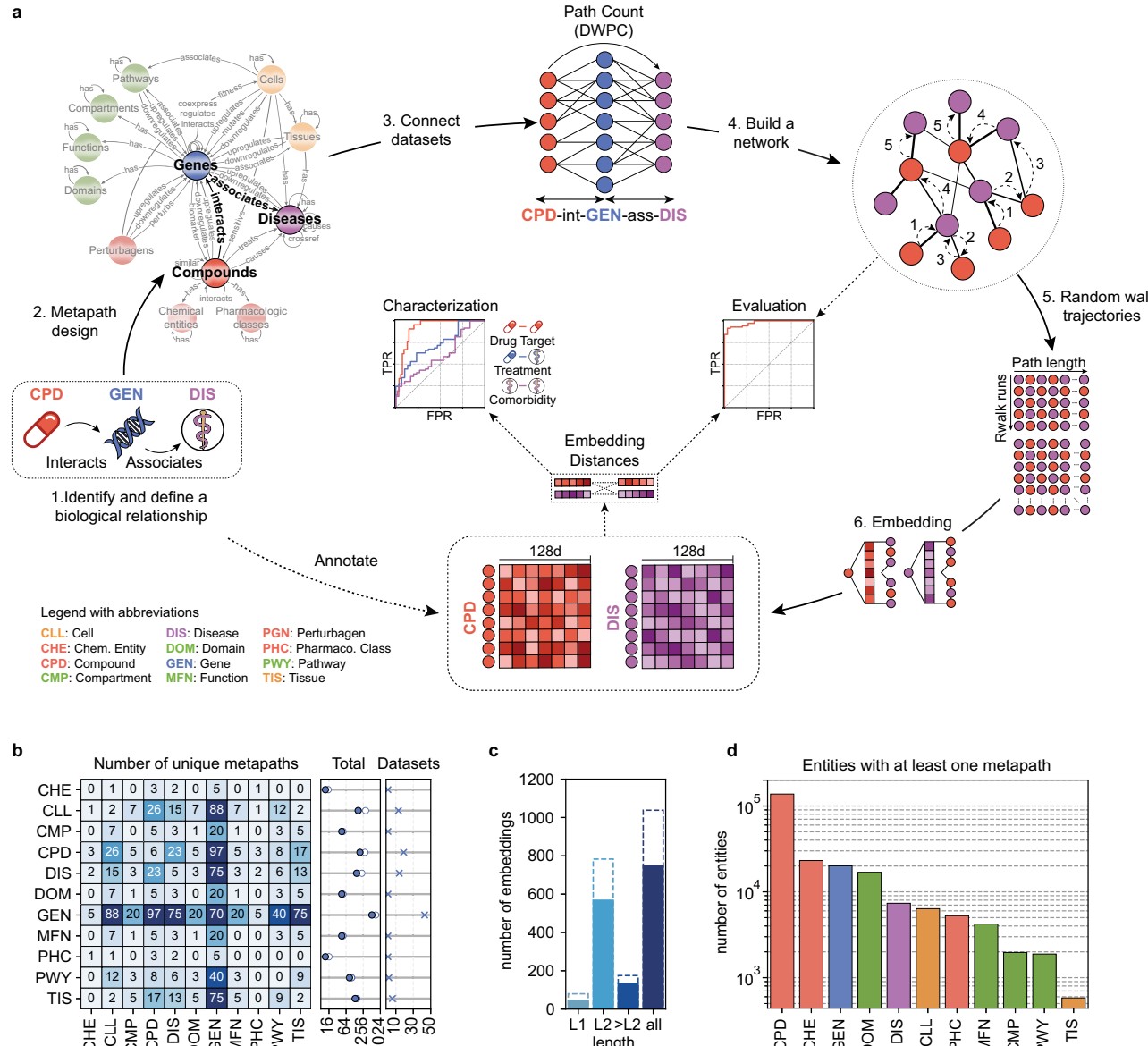

**Fig. 2 | Generating the Bioteque embeddings. a** Scheme of the methodology. First, we define the biological entities to be connected and the specific context to be explored. Then a source-target network is derived by traversing all the paths available from the source to the target nodes of a given metapath. The vicinity of each node in the network is then explored by a random walker and embedded in a 128-dimensional space. Finally, embeddings are evaluated and characterized. **b** Number of unique metapath embeddings linking each entity. In the middle plot, the filled dots indicate the total number of unique metapaths while the empty dots show the total number of metapath-dataset combinations. In the rightmost plot, we show the number of entity-specific datasets used in the metapaths. **c** Number of metapath-dataset embedding combinations obtained at each metapath length. Solid bars highlight the number of unique metapaths. **d** Number of nodes within each entity with at least one embedding in the Bioteque resource. Note that during metapath construction, perturbagen (PGN) entities are always mapped to the corresponding perturbed genes. Thus, although used to construct several metapaths, PGN nodes are not explicitly embedded, i.e., they are not the first or last nodes in the metapaths.

respectively (Fig. 2c and Supplementary Data 3). Length 1 (L1) metapaths correspond to direct associations in the knowledge graph and provide the simplest domain knowledge representations of the entities. Larger metapaths (>L1), on the other hand, are either dedicated to connecting different entities through a third one (i.e., CPD-int-GEN-ass-DIS) or extend L1 associations to similar entities (i.e., CPD-int-GEN-ppi-GEN or CPD-trt-DIS-ass-GEN-ass-DIS), allowing the identification of more complex relationships between biological entities (i.e., two compounds may target different proteins yet affect the same pathway, or CPD-int-GEN-ass-PWY).

Given that the constructed KG is gene-centric, genes (GEN) are the most frequently embedded biological entity in the resource (515

unique metapaths from 43 different datasets) followed by compounds (CPD), cell lines (CLL), and diseases (DIS) (198, 168 and 150 unique metapaths, respectively) (Fig. 2b). Furthermore, most of the metapaths used gene entities, such as those derived from omics experiments or literature curated annotations, as bridges to connect distinct entities (Supplementary Fig. 2). Compounds also play an important role, connecting pharmacological classes and chemical entities to the rest of the graph and being a major source of metapaths embedding cell lines, diseases and tissues.

Overall, the Bioteque provides a collection of 1041 embeddings obtained from 746 unique metapaths, covering all entities defined in the biological KG (Fig. 2d).

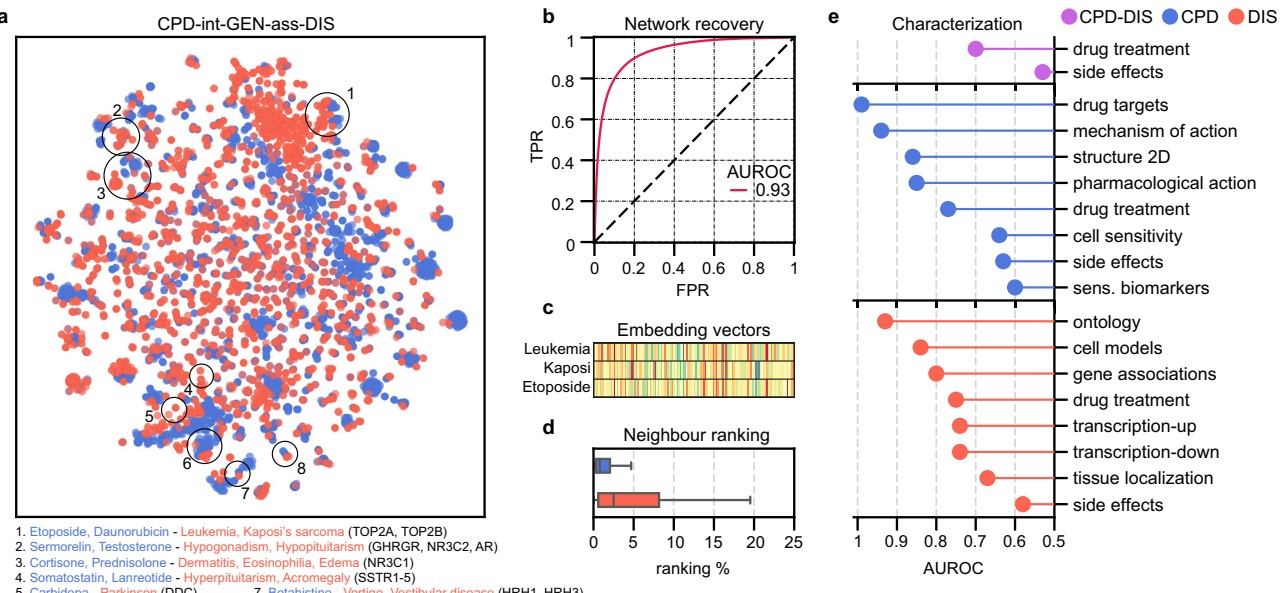

**Fig. 3 | A Bioteque embedding summary card. a** 2D projection (opt-SNE) of the compound (CPD, blue) and disease (DIS, red) embeddings from the metapath 'compound interacts protein associates with disease' (CPD-int-GEN-ass-DIS). We highlight clusters of compounds and diseases sharing treatment evidence. We highlight some representative compounds and diseases found in these clusters, together with the drug targets associated with the diseases. **b** ROC curve validation when reconstructing the original network with the corresponding embeddings. **c** Visual representation of the embedding vectors of leukaemia (top) and Kaposi's sarcoma (middle), together with the drug Etoposide (bottom). **d** Ranking proportion in which the putative CPD ($n = 131,648$) and DIS ($n = 134,997$) neighbours are found. Box plots indicate median (middle line), 25th, 75th percentile (box) and max value within the 1.5*75th percentile (whiskers). **e** Recapitulation of orthogonal associations by using embedding distances. The AUROC (x-axis) summarizes the performance obtained when ranking the orthogonal associations. Drug targets are collected from Drugbank[85], the Drug Repurposing Hub[86] and PharmacoDB[87], and gene-disease associations are obtained from Open Targets[29].

## Embeddings retain the interactions in the original KG, and reveal relationships between biological entities depending on the scope and type of data

Having obtained embeddings for all nodes in the KG, we performed a set of analyses to, on the one hand, validate that the embeddings retained the connectivity observed in the KG and, on the other, to characterize each embedding space in the light of other (orthogonal) datasets in the Bioteque. As an illustrative example, Fig. 3 shows the analysis of the metapath CPD-int-GEN-ass-DIS, corresponding to compounds that interact with genes, which are, in turn, associated with a disease.

To validate the embeddings, we calculated their cosine distances pairwise, and checked that proximal embeddings corresponded to edges in the KG (Fig. 3b), measured with the Area Under the Receiver Operating Characteristic (AUROC) metric. Similarly, when we used the embedding distances to rank entity pairs, we found their known neighbours in the closest 10% of possible nodes (Fig. 3d). Note that the goal of this study is not to benchmark the embedding method (which is already a well-accepted implementation in the field[31]), but to provide an assessment of the approach across a comprehensive set of cases.

Analogously, distances between embeddings can be used to measure whether the dimensional space preserves similarities among entities that share biological traits (i.e., cell lines sharing tissue of origin or genes sharing molecular functions). Following this rationale, we can characterize the type of biological signal captured by a given metapath by comparing its embeddings to a battery of reference biological traits, an approach already used to benchmark drug-drug similarities on the basis of shared chemical features[33]. The use of embeddings allows for straightforward comparison of entities of the same type (for example, similarity of cell lines according to their upregulated genes can be measured by computing distances of CLL entities in the CLL-upr-GEN embedding). Likewise, it is easy to compare and uncover correlations between different types of associations. For instance, the correlation between copy number amplification and upregulation can be assessed by considering similarities in the CLL-cnu-GEN and CLL-upr-GEN embedding spaces. In the CPD-int-GEN-ass-DIS example, drug targets and gene-disease associations are among the biomedical traits that are better recapitulated by the compound and disease embeddings (Fig. 3e). Accordingly, we see how compounds and diseases associated with similar treatments are close in the embedding space. We also observe that compound-disease treatment similarity is achieved at the edge level (AUROC: 0.7), suggesting that not only compounds and diseases with similar treatments are close in the embedding space, but also that compound-disease treatment pairs are often found in the same vicinity. Indeed, compound and disease-associated genes have proven useful in drug treatment prediction exercises[12,34].

A projection of the 128-dimensional embeddings onto a 2D space reveals clusters of drugs and treatments which, by the definition of the metapath, have identifiable targets (Fig. 3a). We find, for instance, drug-disease groups associated with the treatment of leukaemia (e.g., Etoposide and Daunorubicin), hormonal disorders (e.g., Somatostatin and Sermorelin), nervous system disorders (e.g., Carbidopa, Betahistine, and Protriptyline), and inflammatory conditions (e.g., Cortisone and Prednisolone). We observe that most of these drugs target a small subset of proteins or protein families directly related to the diseases, such as the growth hormone-releasing hormone receptor (GHRHR) for hypogonadism treatment, the somatostatin receptor (SSTR) for acromegaly treatment, and the DOPA decarboxylase to prevent dopamine formation in the treatment of Parkinson's disease. Additionally, the analysis reveals that drugs approved to treat either leukaemia or Kaposi's sarcoma cluster, share the topoisomerase II alpha (TOP2A) enzyme as target (Fig. 3c). Indeed, comorbidity between these two diseases has been reported in several studies[35–37], although, to the best of our knowledge, the role of TOP2A in this comorbidity has not been yet described.

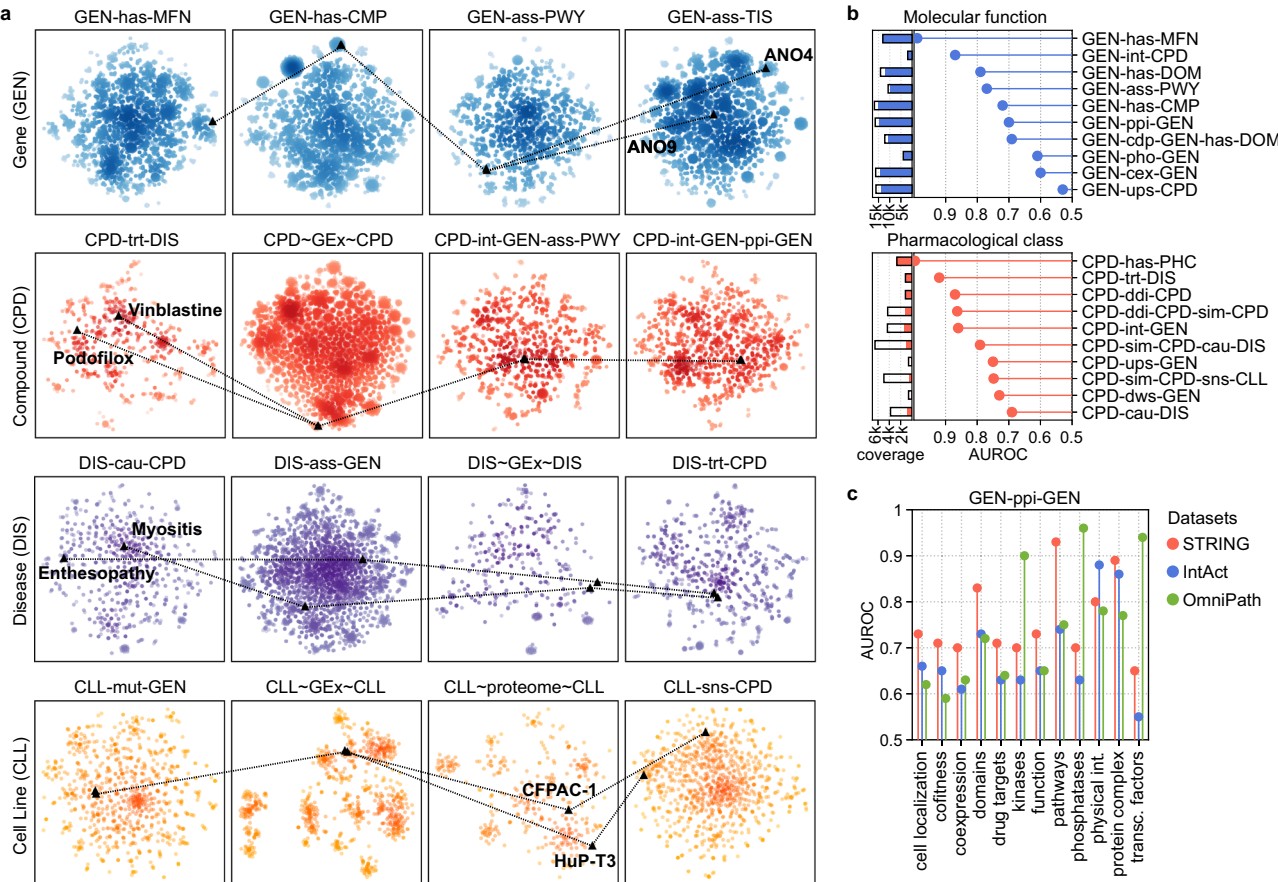

**Fig. 4 | Comparison of embeddings built from different metapaths and datasets. a** Four illustrative examples showing pairs of genes (GEN), compounds (CPD), diseases (DIS) and cell lines (CLL) with similarities or differences depending on the metapaths. The extended nomenclature of each metapath can be found in Supplementary Data 2. **b** Top metapaths (y-axis) recapitulating (AUROC, x-axis) gene molecular function (MFN, blue) and compound pharmacological class (PHC, red). The coloured bars indicate the proportion of nodes in the metapath that could be assessed (i.e., with annotated molecular function or pharmacological classes). **c** Gene embedding characterization of three reference PPI datasets, namely STRING, IntAct and OmniPath. We limited the analysis to the common gene universe (9395 genes) between the three sources.

The repertoire of embeddings encoded in the Bioteque enables exploration of a given biomedical entity from multiple perspectives, often corresponding to different biological contexts, such as genes with the same biological role yet expressed in different tissues, or cell lines with similar transcriptional profiles but dissimilar at the proteome and drug response levels (Fig. 4a). When performed systematically, this analysis quantifies the relationship of a certain metapath with the other metapaths in our collection, which in turn helps assessing the types of biological traits that it captures. Figure 4b shows ten of the top metapaths recapitulating gene molecular function and compound pharmacological class. We see that genes targeted by the same compounds or having similar domains tend to share molecular function while, as expected, sets of interacting compounds, or those with similar binding profiles, tend to belong to the same pharmacological class.

Additionally, one can explore differences among datasets within a single metapath. In Fig. 4c, we embedded three well-known protein-protein interaction (PPI) networks, representing functional interactions (STRING[28]), physical interactions (IntAct[27]), and protein-signalling interactions (OmniPath[38]), and quantified the capacity of these networks to capture a variety of biological features, from cellular localization to protein complexes. The diversity of functional interactions contained in STRING favours recapitulation of most of the features explored, especially those involving similar biological pathways (AUROC: 0.93), protein complexes (AUROC: 0.89) and protein domains (AUROC: 0.83). Not surprisingly, IntAct better preserves

physical interactions (AUROC: 0.88) and shows good performance with protein complexes (AUROC: 0.86). Finally, OmniPath shows an enrichment in signalling processes such as kinase-substrate (AUROC: 0.9), phosphatase-substrate (AUROC: 0.96) and transcription factor interactions (AUROC: 0.94), in good agreement with the type of resources used to build this network.

In general, the different considerations followed to populate these networks may favour some domains of knowledge, hence suiting different tasks, which can be efficiently and systematically revealed by transforming them into embeddings. In the next sections, we present three illustrative examples on how these biological embeddings can be used off-the-shelf in a variety of tasks.

## Gene expression embeddings as biological descriptors of cell lines

Gene Expression (GEx) experiments have been widely used to characterize cellular identity and state, as they broadly recapitulate tissues of origin[39] and they are notable genomic biomarkers for anticipating drug response[40]. However, these experiments typically measure the expression of 15–20k genes, yielding numerical profiles that are computationally demanding and prone to overfitting problems when used as input in machine learning approaches with limited data[41,42].

We thus explored whether our more succinct 128-dimensional vectors were able to retain the information contained within the full GEx profile. Taking the Genomics of Drug Sensitivity in Cancer (GDSC)[40] panel as a reference, we collected, for each cell line, the basal

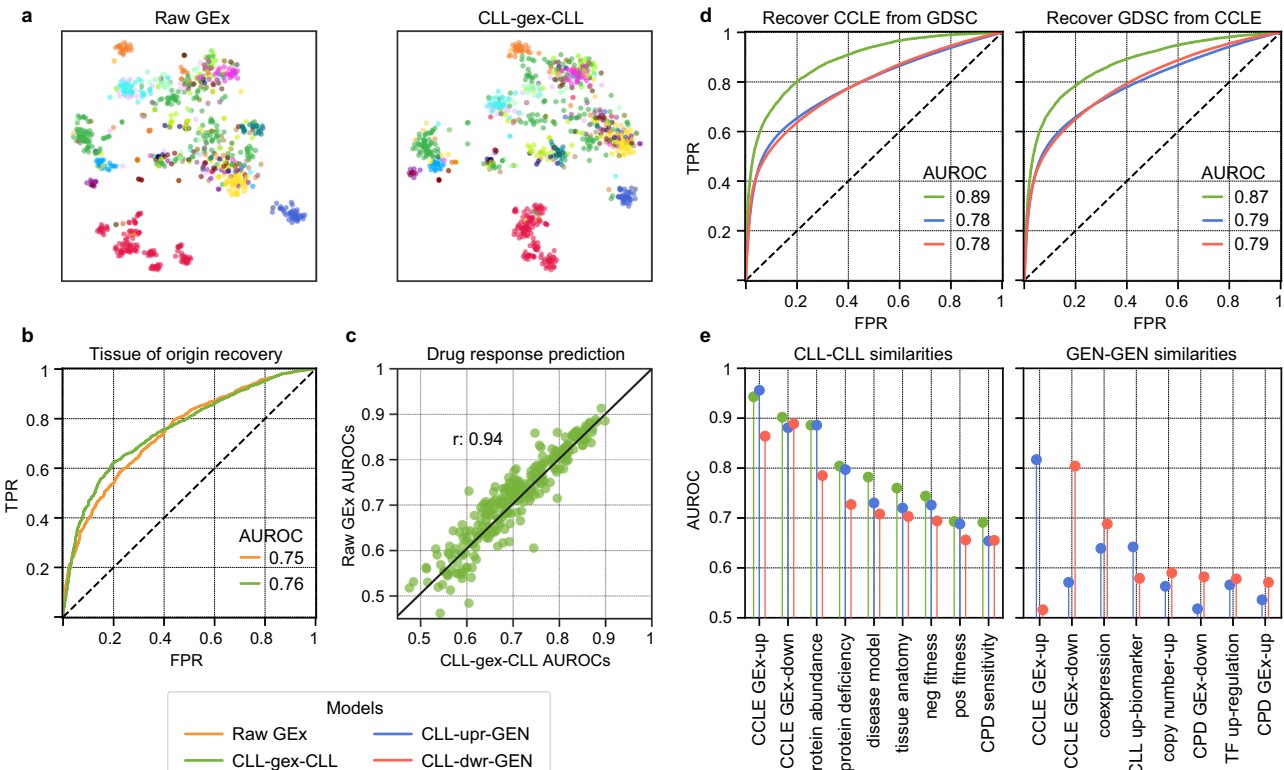

**Fig. 5 | Analysis of gene expression (GEx) embeddings. a** 2D projection of the raw GEx (left) and the corresponding Bioteque 'cell has similar gex cell' (CLL-gex-CLL) embedding (right). Each dot corresponds to one cell line and is coloured by tissue of origin. **b** Tissue recovery by the raw GEx and the CLL-gex-CLL embedding. **c** Drug response prediction performance (AUROC) for each drug in the GDSC resource from models trained with either the raw GEx (y-axis) or the CLL-gex-CLL embeddings (x-axis). **d** Recovering CCLE (left) and GDSC (right) cell-cell (CLL-CLL) similarities (green), cell-gene (CLL-GEN) upregulation (upr) similarities (blue) and CLL-GEN downregulation (dwr) similarities (red) using embedding distances from the GDSC and the CCLE embedding spaces, respectively. **e** Characterization of the CLL-CLL (left) and GEN–GEN (right) embedding similarities for three metapaths: CLL-gex-CLL (green), CLL-upr-GEN (blue) and CLL-dwr-GEN (red).

(raw) GEx (17.7 K Genes) and the corresponding Bioteque metapath embedding CLL-dwr+upr-GEN-dwr+upr-CLL (hereafter CLL-gex-CLL), aimed at capturing gene expression similarities between cell lines.

We first examined the similarity landscape of the cell lines by performing a 2D projection of the raw and embedded GEx. By colouring the cell lines according to their tissue of origin, we visually verified the capacity of the CLL-gex-CLL embedding to resemble the raw GEx data (Fig. 5a). Indeed, cosine similarities between CLL-gex-CLL vectors up-ranked CLLs sharing tissue of origin with a similar rate as when using correlations between raw GEx vectors (AUROC: 0.75 and 0.76, respectively) (Fig. 5b).

Next, we assessed the capacity of our embeddings to predict the drug response of each cell line. To this end, we trained a standard machine learning model (a random forest classifier) for each of the 262 drugs in the panel and predicted sensitive/resistant responses using the raw GEx and our embeddings independently ("Methods"). Indeed, we found that the capacity of the CLL-gex-CLL embedding to recapitulate drug response is equivalent to that observed when the raw GEx data is used (average AUROC: 0.70 and 0.71, respectively). Moreover, the models based on embeddings had strong concordance with the raw GEx model (0.94 Pearson correlation) (Fig. 5c). This level of agreement is remarkable and represents a clear advantage for the embeddings since they are smaller, easier to handle and do not require expert knowledge to pre-process the raw data. A disadvantage of the embedding approach is the less obvious interpretability of predictions.

After verifying that the Bioteque GEx embeddings retain the basal transcriptional information from the cell lines, we used them to compare profiles obtained from different cell line panels. Specifically, we compared the GDSC with the Cancer Cell Line Encyclopaedia (CCLE)[43]. In agreement with previous reports, we observed a strong correspondence between the two panels, measured as CLL-gex-CLL similarities in the embedding space (AUROC: 0.89) (Fig. 5d). To assess whether these similarities were driven by the up- or downregulation of the same genes, we repeated the analysis focusing on the CLL-upr-GEN and CLL-dwr-GEN embeddings and checked whether the CLL-GEN similarities in the GDSC panel were also preserved in the CCLE. In general, the recovery score of cell line-specific up-/downregulated genes (i.e., CLL-GEN pairs) was lower (AUROC: 0.78) (Fig. 5d). We obtained similar results when we reversed the exercise and used CCLE embeddings to recapitulate GDSC similarities (Supplementary Fig. 3). This finding suggests that, while cell line similarities between panels are robust (i.e., cell lines sharing similar transcriptional signatures in one panel also share similar ones in the other), the specific transcriptional changes of a given cell line may differ. The characterization of the CLL-CLL and GEN–GEN distances further confirmed the better recapitulation of cell line similarity in comparison to gene similarity between panels (AUROC: 0.9 and 0.8 for the CLL-CLL and GEN–GEN similarities, respectively) (Fig. 5e). Furthermore, the CLL-CLL similarity characterization revealed a strong concordance between protein and transcript levels (AUROCs: 0.9 and 0.8 for protein abundance and deficiency, respectively), which was partially driven by the same CLL-GEN pairs (AUROC: 0.72 and 0.63 for the protein abundance and protein deficiency CLL-GEN pairs, respectively) (Supplementary Fig. 3). In addition to tissue of origin, we also observed resemblances between cell lines used to model a given disease (AUROC: 0.78), sharing fitness profiles (AUROC: 0.72 for negative and 0.69 for positive fitness profiles) and similar drug responses (AUROC: 0.7). Finally, the GEN–GEN

similarities also revealed a mild recapitulation of known co-expressed gene pairs (AUROC: 0.64 and 0.69, for the up- and downregulated gene similarities, respectively), thereby suggesting that some of the genes commonly up- or downregulated in the same cell lines from different panels may share the same transcriptional regulatory programmes.

On the whole, our approach retains meaningful information from the original data into a reduced number of dimensions (128 vs ~20k), even when the data comes from a much noisier source such as transcriptomic technologies. We believe that the standardized and dense format of our embeddings provides a by-default way to integrate and compare omics datasets.

## Assessing the uniqueness of new omics datasets

Since the consolidation of high-throughput omics technologies, several long-term initiatives have been established to comprehensively characterize certain levels of biological systems (i.e., genetic interactions in yeast[44] or the transcriptomes of cell line panels and human tissues[43,45]). After several years running, all these efforts have had to balance a potential decrease in novelty and an increase in costs as the screens approach saturation. The Bioteque provides a corpus of biological data that is cast to a single format and, as such, it offers a means to quantify the degree of novelty of new data releases of omics experiments. As an illustrative example, we analyse the systematic charting of the Human Reference Interactome (HuRI) with the yeast two-hybrid methodology, which has already identified over 50,000 protein-protein interactions (PPIs) of high quality over the last 15 years[46–48].

To estimate the level of support from different experiments and assess the novelty of the latest HuRI release (HuRI-III[48]), we used the embedding space of relevant metapaths to determine the biological context of each pair of interacting proteins. In brief, for each gene-gene pair, we calculated an empirical *P* value corresponding to the measured similarity in the embedding space, which allowed for commensurate comparison of distance/similarity measures performed in different embedding spaces (see "Methods"). Note that, to have a fair representation of the known physical interactions, we embedded an older version of the protein-interaction network, without including any of the entries from HuRI-III. We then categorized each interaction in HuRI-III into four groups, depending on the level of support contained in the Bioteque embeddings. In this regard, we labelled them as (i) known and supported interactions (covered by GEN-ppi-GEN and at least another metapath), (ii) known interactions (only covered by GEN-ppi-GEN), (iii) supported interactions (covered by other metapaths but not GEN-ppi-GEN) and (iv) potentially novel interactions (with no apparent support in any of the metapaths screened) (Fig. 6a). Remarkably, after three updated versions of HuRI, almost half of the interactions can be classified as potentially novel according to the selected metapaths. Moreover, although only 5825 (11%) of the interactions were supported by GEN-ppi-GEN embeddings, mostly coming from previous versions of HuRI[46,47], our analysis suggests that a higher proportion can be recovered. In fact, at 0.05 FDR ("Methods"), the GEN-ppi-GEN embedding recovered 18% of HuRI-III, retrieving 5456 (94%) of previously known interactions while finding 3994 new pairs (Fig. 6b). On the other hand, we observed a substantial number of physical interactions presumably involved in similar pathways (GEN-ass-PWY), cellular components (GEN-has-CMP), or protein domains (GEN-has-DOM). At 0.05 FDR, these metapaths alone recovered 6905 unique interactions of which 4484 (65%) were not obvious from the physical interaction space (Fig. 6c). To delve into the correlation and relative importance of the metapath for explaining PPIs, we used the *P* values as features for a tree-based machine learning model trained to identify HuRI-III edges. We then assessed the importance of each metapath for the prediction using Shapley values[49]. As visually anticipated from the heatmap, the model achieved a reasonable performance (AUROC: 0.69), mostly relying on previously known physical interactions, cellular components, protein domains, and pathways, all of them showing a certain degree of agreement (Supplementary Fig. 4). Interestingly, we also identified successfully predicted cases with little to no evidence from physical PPIs. For instance, our metapath distance-based model predicted the interaction between the neuronal proteins HOMER1 and SHANK2, the tRNA-splicing endonuclease TSEN54 and the polyribonucleotide CLP1, and the adenosine deaminase ADARB1 and the protein kinase PRKRA, none of which had any reported evidence in protein interaction databases but showed strong positive support in the GEN-ass-PWY, GEN-has-CMP, and GEN-has-DOM metapaths, respectively (Fig. 6d). Indeed, some of these associations have been related in other contexts[50–52], but with no indication of physical interactions before HuRI-III.

We have shown how the continuous and interpretable dimensional space of the Bioteque embeddings provides a powerful framework for characterizing individual observations, which can, in turn, be exploited to guide the interpretation of the entire dataset and, to some extent, assess the novelty of the data.

## Discovery of drug repurposing opportunities using the multiple scopes offered by the embeddings

Drug repurposing is often regarded as an attractive opportunity to quickly develop new therapies[53]. However, perhaps with the exception of cancer, where abundant models and molecular data are available, it is difficult to generate data-driven predictors to suggest new uses for approved or investigational drugs, mainly due to the lack of disease descriptors and the small number of known drug-disease indications. Indeed, according to the last update of repoDB, half of the drugs (1097) have only one approved indication, and a third of the diseases (458) are treated with only one drug (Supplementary Fig. 5). Thus, training models with all the known drug-disease associations and later transfer of the insights gained to underexplored treatment areas would be highly desirable[54,55].

To explore whether the Bioteque could be useful in this scenario, we set out to predict new compound-disease indication pairs introduced in repoDB in 2020 (v2) training a model on the previous version (v1), launched in 2017 ("Methods"). We mapped all disease terms to the Disease Ontology, removed redundant indications (according to the ontology), and trained a conventional random forest classifier to predict whether a given CPD-DIS corresponds to a true therapeutic indication. We used two sets of metapath embeddings: one in which we used L1 metapaths (*Short*) based on the drug targets (CPD-int-GEN) and gene associations (DIS-ass-GEN), and another in which we used L3 metapath (*Long*) linking the pharmacological class and the treatment of known CPD and DIS to those sharing drug target (CPD-int-GEN-int-CPD-has-PHC) or gene associations (DIS-ass-GEN-ass-DIS-trt-CPD), respectively. We chose to use drug targets and gene associations because we observed that their embeddings broadly recapitulate the pharmacological class and the disease treatment for a sufficient number of nodes (Supplementary Fig. 5). Moreover, to assess the capacity of the gene-based similarities to correctly infer the treatment, we also tested a metapath (*Long-b*) in which we prevented the CPDs and DISs from being linked, thus making the association with PHC or treatment purely based on the gene-driven similarity to other CPD or DIS. To avoid trivial predictions, we removed associations with PHCs or treatments for drugs and disease unique to the repoDB v2 in all *Long* metapaths. As a basal model, we used chemical fingerprints (ECFP4, 2048 bits) for the CPDs and either one-hot identity vectors (*Basal1*) or binary gene annotations (*Basal2*) for the DISs.

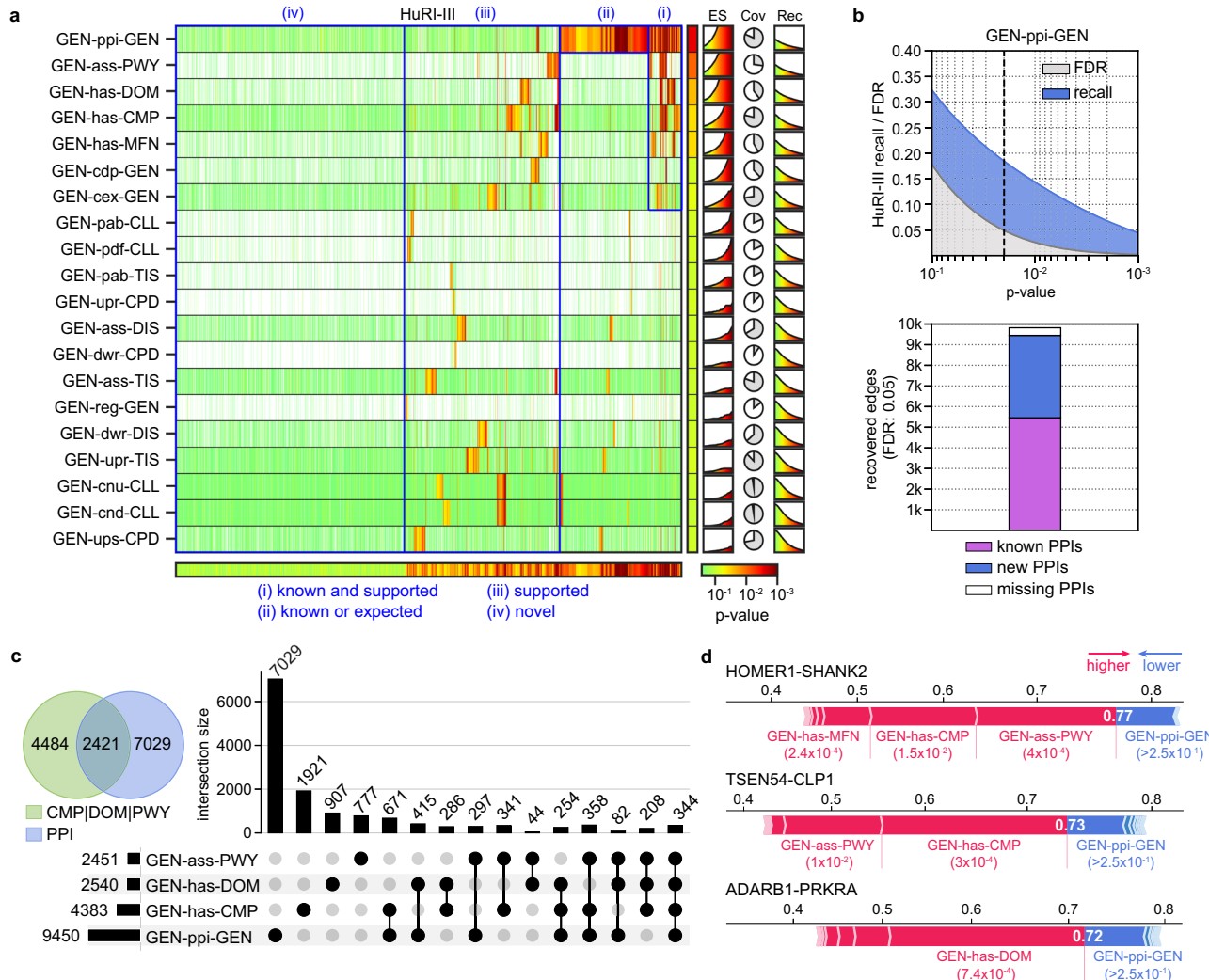

**Fig. 6 | Assessing the novelty of the HuRI-III interactome. a** Embedding distance *P* values are calculated for each PPI in HuRI-III (x-axis) using the corresponding gene-gene (GEN–GEN) embeddings from a subset of metapaths (y-axis). Please, note that these *P* values do not reflect the significance of any statistical test, but indicate the normalized quantile rank position of a given observation in a background distance distribution ("Methods"). Red tones (lower *P* values) indicate similarity according to a given embedding space. The column and row next to the heatmap show the 10th percentile of the *P* value distribution for each metapath and the lowest *P* value for each edge, respectively. In blue, we grouped edges according to four levels of support. On the right, it is shown the enrichment scores (ES) (capped between 1 and 5 on the y-axis) across *P* values, the coverage (Cov), and the cumulative recall (Rec) across *P* values. **b** (Top) Recovery of HuRI-III edges (recall) and randomly permuted edges (FDR) by 'protein interacts protein' (GEN-ppi-GEN) embeddings across the *P* values (x-axis). The dashed line is placed at the 0.05 FDR (corresponding to a *P* value of 0.02). (Bottom) Number of HuRI-III interactions

recovered by the GEN-ppi-GEN embedding at 0.05 FDR stratified by those covered in the original network (known PPIs), those not available in the network, hence, predicted by the embeddings (new PPIs), and those present in the original network but not covered at the given *P* value (missing PPIs). **c** Number of unique HuRI-III edges recovered at 0.05 FDR by the GEN-ppi-GEN and/or the three most supportive metapaths, including 'gene has cellular components' (GEN-has-CMP), 'protein has domain' (GEN-has-DOM), and 'gene associates with pathway' (GEN-ass-PWY). **d** Shapley force plots corresponding to the prediction of three PPIs with no direct evidence of physical interaction before HuRI-III was released. Red segments are metapath-specific *P* values that pushed predictions toward a high probability of interactions, while blue segments pulled predictions towards a low probability. The length of the segments is proportional to their impact on the prediction. The final output probability given by the model is found where both forces equalize (shown in white).

We considered two use cases: a drug repurposing exercise, in which we ranked all the diseases predicted to be potentially treated with a given compound, and a prescription exercise, in which we ranked all compounds that might be useful to treat a given disease. In both scenarios, the three metapath embeddings showed remarkable predictive power compared to the basal models, with the model built from *Long* embeddings being the one with superior performance (Fig. 7a). Specifically, for half the tested compounds, the *Long* embeddings model found a new validated therapeutic purpose within the top 2% of disease predictions (corresponding to the top 10 ranked diseases). Analogously, for roughly 50% of the diseases, the model found a

correct treatment within the top 1% of compound predictions (corresponding to the top 8 ranked compounds). Furthermore, although with poorer performance, our biological embeddings were able to yield correct predictions for compounds and diseases with minimal evidence available (i.e., with only one known indication or treatment in repoDB v1) (Fig. 7a, dotted lines). In contrast, the best performing basal model (*Basal2*) found correct predictions for 32% of the compounds and 41% of the diseases within the same ranking range. Moreover, the Bioteque-based models were better at consistently up-ranking indications (or treatments) of compounds (or diseases) with multiple new annotations in repoDB v2 (Fig. 7b). In fact, among our top

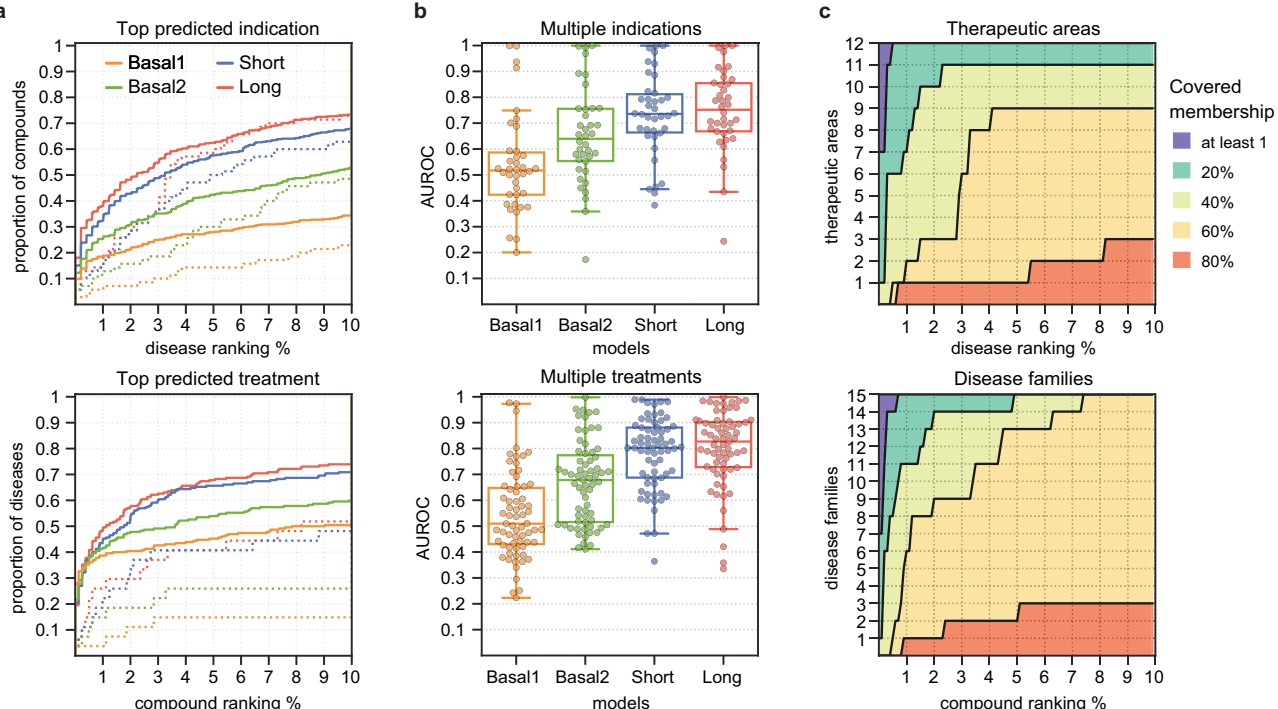

**Fig. 7 | Prediction of drug indications and disease treatments from repoDB.**
**a** Cumulative distribution (y-axis) of compounds (top) and diseases (bottom) according to the ranked position (x-axis) of the top predicted disease indication (top) or compound treatment (bottom) for the four tested models. The rankings are shown in percentages and only for the first 10% of compound/disease predictions (corresponding to the top 50 and 80 diseases and compounds, respectively). Dotted lines show the distribution for those compounds or diseases with only one positive indication in repoDB v1. **b** Classification performance obtained for each compound (n = 38, top plot) and disease (n = 67, bottom plot) with multiple (≥5) new indications reported in repoDB v2. Box plots indicate median (middle line), 25th, 75th percentile (box), and max and min value within the 1.5*25th and 1.5*75th percentile range (whiskers). **c** Number of different therapeutic areas (top) and disease families (bottom) covered by the predictions of the *Long* model. We considered a given therapeutic area or disease family to be covered when the model predicted one true indication or treatment (as in panel (**a**)) for at least 1%, 20%, 40%, 60%, or 80% of its instances.

predictions, we found repurposing cases that reached clinical trials (Supplementary Fig. 6a). For instance, while both Verapamil and Ranolazine drugs have been approved for the treatment of angina pectoris, our model correctly predicted the repurposing effect of Verapamil in the treatment of ischaemic stroke (clinical trial: NCT02823106) and Ranolazine in the treatment of atrial fibrillation (clinical trial: NCT03162120) in the top 1 and 2 positions, respectively (Supplementary Fig. 6b). Interestingly, our model highlights hyperinsulinemia as the top repurposing for Ranolazine. While this link is not included in repoDB, we have found diverse studies supporting the correlation of Ranolazine with insulin levels[56–58]. Finally, we verified that these predictions covered a broad range of therapeutic areas and disease families. Indeed, we found that within the top 1% of predictions, the *Long* model successfully predicted one indication or treatment for 20% of all the compounds and diseases in each therapeutic area or disease family (Fig. 7c and Supplementary Fig. 6e). These results were reproduced with the *Long-b* model, showing that, as expected, the genes associated with drugs or diseases of known treatment can indeed be used to better infer the activity of drugs and diseases with unknown indication (Supplementary Fig. 6c, d).

Overall, we showed how Bioteque embeddings can be directly plugged into machine learning models, and how, by combining different context associations into larger metapaths, they can increase the performance of drug-disease prediction models. Indeed, we used a preliminary version of Bioteque embeddings to successfully identify potential targets for a set of kinase inhibitors from perturbational profiles, including drug-induced transcriptional changes and cell sensitivity data, in several cell lines[59].

## The Bioteque resource

We built an online resource to facilitate access to all the pre-calculated Bioteque embeddings (https://bioteque.irbbarcelona.org). The Bioteque web offers a visual way to explore over one thousand metapaths by selecting the nodes to connect, as well as the type of relationship between them. For a selected metapath, we provide an analytical card displaying a 2D representation of the embedding, a ROC curve assessing the preservation of the original network, distance distributions of the embedding space, and biological associations that are best recapitulated by the metapath of interest.

Furthermore, the web page also offers a section were metapath embeddings and other metadata can be downloaded. The generated file contains the embeddings for each node, the nearest neighbours of each node in the space, and the analytical card displayed on the web. Additionally, we make available executable notebooks showing how to download our embedding resource programmatically as well as how to perform most of the downstream analyses presented throughout this manuscript. More specifically, we illustrate how to (i) generate 2D (interactive) visualizations that can be coloured and annotated according to side information (e.g., colour cell lines by tissue of origin), (ii) identify similar nodes (close neighbours) for a given entity of interest, (iii) cluster the embedding space and (iv) build a predictor model trained on our embeddings.

The Bioteque web also provides information on the specific sources used to construct each metapath, and some general statistics on the contents of the current version of this web resource. We also provide a link to our GitLab repository, which contain the full code necessary to pre-process the data to generate and analyse biological embeddings (http://gitlabsbnb.irbbarcelona.org/bioteque). The entire

resource, including the underlying data and biological embeddings, will be updated once per year, or as soon as a major dataset is released.

## Discussion

With the accumulation of large-scale molecular and cell biology datasets, coming from ever-growing literature, omics experiments and high-throughput screenings, new frameworks for integrative data analysis are necessary. For a given biological entity (e.g., a gene), we are now able to stack multiple layers of its biological complexity (e.g., its structure, function, regulation, or interactions), which offers an opportunity for a more complete, systemic view of biological phenomena, but brings along several challenges, including the handling of different data structures, nomenclatures, signal strengths, and variable dimensionalities.

To tackle these challenges, we have developed the Bioteque, a resource of pre-calculated, fixed-format vector embeddings built from a comprehensive biomedical knowledge graph (KG). The KG contains physical entities like genes, cell lines, and compounds, as well as concepts like pathways, molecular functions, and pharmacological classes. Embeddings capture the connections between nodes in the KG according to a certain metapath, i.e., a sequence of semantic and/or mechanistic relationships between entities. We have shown how this approach is useful to (i) produce compact descriptors that broadly preserve the original data, (ii) systematically characterize biological datasets such as cancer cell line transcriptional signatures, (iii) assess the novelty of a given omics experiment and (iv) mine for drug repurposing opportunities based on multiple associations between drugs and diseases.

In the Bioteque, we have incorporated datasets from over 150 distinct sources, keeping the integrity of the original data to a feasible extent and applying standard transformations when required. Note that the accuracy of the Bioteque is determined by the quality of the source data. As experimental technologies continue to evolve, new information will populate these databases and novel standards will emerge, opening the door for more comprehensive and higher quality embeddings. In addition, as a first attempt, we used a network embedding technique that purely relies on the graph topology built from the biomedical data, in contrast to other techniques that also leverage node and edge attributes (e.g., Graph Neural Networks, GNN). While these methods may contribute to improving the embedding space, their quality depends on the availability of enough data and meaningful node features, while requiring a thorough fine-tuning of the hyperparameters[60,61]. Taken together, the proper implementation of these methods becomes unfeasible for the systematic embedding of thousands of networks. Additionally, the incorporation of external node features in the network could compromise the controlled identity of the metapaths. Nevertheless, Bioteque descriptors can be easily recycled as node features for new task-specific networks, thus transferring the learning encoded from orthogonal biomedical datasets to more complex, attribute-aware models. Finally, we would like to point out that there are parts of the current biomedical knowledge that have not yet been included in the resource, such as antibody-target interactions and metabolomics. As a molecular/cell-centric resource, the Bioteque also lacks patient-derived data[13], including interactions with the microbiome[62]. Updated versions of the Bioteque will have to be complemented with the incorporation of other fields of biological knowledge, the re-accommodation of the datasets in the resource (based on updated standards), and the improvement of embedding strategies to account for side-features of the nodes or incorporate unseen (external) nodes in the embedding space. Moreover, future developments will explore the adoption of biological descriptors as features for a variety of downstream-specific tasks, including a systematic screening of the biological support of wet lab experiments or the modelling of complex diseases to guide the generation of new chemical entities to tackle them[20].

## Methods

### Building the metagraph

All gathered data was stored in a graph database (KG) in which nodes represent biological or chemical entities and edges represent associations between them.

**Nodes (entities).** The nodes in the graph can belong to one of 12 types (aka metanodes). For each entity type, we predefined a universe of nodes and chose a reference vocabulary based on standard terminologies. These 12 entity types are (in alphabetical order):

**Chemical entities (CHE).** Chemistry terminologies extracted from the Chemical Entities of Biological Interest (ChEBI) ontology[63].

**Cells (CLL).** Cell lines used in biomedical research and extracted from the Cellosaurus resource[64].

**Cellular Components (CMP).** Biomolecular structures and complexes as defined by the Gene Ontology[65] (extracted from the basic filtered ontology).

**Compounds (CPD).** Small molecules codified with the standard InChIKey. As we do not use any predefined library of compounds, the universe will be determined by the union of compounds included in other datasets (e.g., drug–target interactions).

**Diseases (DIS).** Abnormal conditions, drug side effects and symptoms. We used the Disease Ontology[66] as a reference vocabulary.

**Domains (DOM).** Functional and structural protein domains extracted from InterPro[67].

**Genes and proteins (GEN).** Genes and proteins were unified and stored by Uniprot[68] accession code (UniProtAC). We worked on the reviewed Human proteome.

**Molecular functions (MFN).** Biological function of the proteins defined by the basic Gene Ontology[65].

**Perturbagens (PGN).** CRISPR, overexpression, and shRNA perturbations. Note that PGNs are always mapped to the corresponding perturbed gene when constructing the metapath. Therefore, instead of providing PGN labels, we provide the UniProtAC of the perturbed genes.

**Pharmacologic classes (PHC).** Pharmacologic classes defined by the Anatomical Therapeutic Chemical (ATC) code (http://www.whocc.no).

**Pathways (PWY).** Biological pathways and processes. We used Reactome[69] as a reference vocabulary.

**Tissues (TIS).** Anatomical tissues and cell types defined by the BRENDA Tissue Ontology[70].

Please note that in the datasets containing ontological terms (CMP, DIS, MFN and PWY), we removed the least informative terms (i.e., those that are higher up in the ontology). These terms were identified by calculating the information content[71]. The node universe for each entity and the list of removed terms are available in Supplementary Data 1.

**Vocabulary mapping.** To integrate terminologies, we extracted curated cross-references from the official terminology sources and associated ontologies. As the nomenclatures used to identify diseases and pathways were particularly diverse and rarely cross-referenced, we further increased the mapping of these terms by inferring similarities within concepts as detailed below.

Diseases were mapped by calculating disease term similarities through shared cross-references to the Unified Medical Language System (UMLS), obtained from the DisGeNET mapping resources (https://www.disgenet.org/downloads). Specifically, we encoded each disease term into a binary vector spanning the universe of UMLS terms of all nomenclatures. We then transformed the binary vectors with the corresponding term frequency-inverse document frequency (TF-IDF) values and computed pairwise cosine distances between the Disease Ontology and the rest of the vocabularies. Using the similarities obtained from curated cross-references as reference, we found a cosine similarity cutoff of 0.5 to correspond to an empirical $P$ value of $5 \times 10^{-4}$.

Pathway cross-references were extracted from the ComPath resource[72] and extended following the PathCards[73] approach. This approach first clusters the pathways into SuperPaths based on overlapping genes and then uses Jaccard similarities between the Super-Paths genes to define pathway similarity. We used the same parameters described in the PathCards paper (0.9 for the overlap cutoff, 20 minimum genes in the pathways, and a Jaccard similarity of at least 0.7).

**Edges (associations).** Edges in the graph are used to link biological and/or chemical entities. Since two entities may be connected by multiple edge types (i.e., 'compound treats disease' or 'compound causes disease'), we define the associations as triplets (metapaths) of entity-relationship-entity (CPD-trt-DIS, CPD-cau-DIS).

Homogeneous associations are those concerning entities (metanodes) of the same type (e.g., 'gene is co-expressed with gene', GEN-cex-GEN), while heterogeneous associations are related to entities of different types (e.g., 'tissue has cell', TIS-has-CLL). Note that we annotated only one direction of the heterogeneous associations (in fact, we kept CLL-has-TIS instead of TIS-has-CLL), although both directions are valid when defining metapaths. On the other hand, edges were treated as directional whenever a homogeneous association had only one valid directionality, like in the case of kinase-substrate interactions ('gene phosphorylates gene', GEN-pho-GEN) or transcription factor regulations ('gene regulates gene', GEN-reg-GEN). Finally, edges corresponding to similarity measures required a predefined set of nodes for pairwise comparison, and they were computed only after the rest of the graph was populated.

### Populating the knowledge graph with data

For each type of association or metaedge, we can have one or more datasets (Supplementary Data 2). Datasets are not merged but kept as individual sources so that they can be embedded individually or in combination within a given metapath. The dataset processing pipeline consisted of two steps. In the first step, nomenclatures were standardized and cutoffs were applied. In the second, applied only to ontological data, terminologies were mapped and the network was pruned.

**Dataset standardization.** We processed each dataset individually in order to handle the diversity of formats and data types. The guiding principles of data processing were those defined by the Harmonizome[7].

Datasets that already provided binary data were integrated naturally by converting them into the network format of the KG. If the database provided a measure of confidence (e.g., edge weights or $P$ values), we applied default cutoffs (if given) and/or followed author recommendations in order to remove spurious interactions. To build the network, we did not use any edge weight coming from the original source during the embedding process. This was motivated by the observation that most of these weights are based on a measure of support or confidence, which does not necessarily reflect biological significance/strength. Instead, these scores usually capture biases on the knowledge annotation (e.g., associations for under-studied diseases will be less covered among the different sources and, therefore, are prone to have lower confidence scores) or detectability limitations of the experimental screening (e.g., the abundance level of some proteins are more difficult to detect than others). While weighted edges could provide valuable information for the embedding, we could not find a general way to treat them across the diverse and heterogeneous associations in our resource.

Occasionally, the same dataset can be further divided into different subsets on the basis of a given categorical variable (e.g., curated/inferred). We kept these subsets as independent datasets when applicable. For instance, there is a curated version of DisGeNET and an inferred version of it.

Continuous data requires the application of a cutoff before its integration in the KG. Below, we detail how these cutoffs were chosen depending on the nature of the data.

**Transcriptomics and proteomics data.** We adapted the strategy followed by Harmonizome, which is based on traditional statistical treatment of gene expression profiles. More specifically, we first mapped the samples and genes to our reference vocabulary and collapsed the duplicates by their mean value. A log2 transformation was then applied followed by a quantile normalization of the genes (unless the dataset was already transformed by the data providers). Next, we subtracted the median and scaled the data according to the quantile range of each gene. Finally, the top 250 most positive and negative genes were selected for each sample and kept in the corresponding metaedges (e.g., CLL-upr-GEN and CLL-dwr-GEN).

**Drug sensitivity.** To binarize drug sensitivity data, we used the waterfall method first described by Barretina et al.[74], and used since then in different subsequent works (e.g.,[75–77]). This method ranks cell lines on the basis of a drug response measure, for instance, the area under the growth inhibition curve (AUC), and uses the shape of the plot to define a sensitivity threshold. The waterfall method was applied for each compound in the dataset, keeping at least 1% but no more than 20% of sensitive cell lines and requiring an AUC sensitivity value lower than 0.9.

**Perturbation experiments.** Gene perturbation data required a preliminary step to differentiate the type of perturbation (e.g., 'CRISPR modification silences gene A') from its outcome (e.g., 'silencing gene A results in overexpression of gene B'). First, for each perturbation in the dataset, we created a perturbagen (PGN) node with a unique identifier. We then simplified the two-step relationship (e.g., 'perturbagen that silences gene A upregulates gene B') into a 'perturbagen upregulates gene B' association (PGN-upr-GEN).

**Other datasets.** For some datasets containing continuous data, we had to apply customized approaches to convert them into a network format. Details about the pre-processing of each particular dataset are provided in Supplementary Data 2, while the corresponding Python scripts can be found on https://bioteque.irbbarcelona.org/sources.

**Terminologies and pruning.** Six terminologies (namely, CMP, DOM, MFN, PHC and PWY) had semantic relationships between them. In these cases, we propagated all the reported relationships with other terms (e.g., GEN) through the parents of their corresponding

ontologies. To maximize coverage, propagation was done before cross-referencing.

## Selection of metapaths

We chose a controlled set of metapaths for which we pre-computed embeddings. These are the embeddings that are deposited in the Bioteque resource. The metapaths were selected as follows.

**Length 1 (L1)**. All possible metapaths of length 1 are embedded except for those capturing cross-references (DIS-xrf-DIS), ontologies (PWY-hsp-PWY), compound-compound similarities (CPD-sim-CPD), and PGN associations. Note that PGN nodes are mapped to the corresponding perturbed genes through the PGN-pdw-GEN or PGN-pup-GEN metapaths (thus, >L1 metapaths).

**Length 2 (L2)**. Only the mimicking (e.g., CLL-dwr+upr-GEN-dwr+upr-CLL) or reversion (CLL-upr+dwr-GEN-dwr+upr-CLL) of both directions (up/down) are used for metapaths connecting entities through transcriptomic, proteomic or transcription factor signatures. CLL and TIS are always connected through the CLL-has-TIS association. Finally, only the following associations are allowed when linking cells and genes within a metapath: CLL-upr-GEN, CLL-dwr-GEN, CLL-mut-GEN.

**Length 3 (L3)**. L3 metapaths are constructed by linking L1 metapaths with any of the following L2 metapaths: CLL-dwr+upr-GEN-dwr+upr-CLL; CLL-has-TIS-has-CLL; CMP-has-GEN-has-CMP; CPD-has-PHC-has-CPD; CPD-int-GEN-int-CPD; DIS-ass-GEN-ass-DIS; DOM-has-GEN-has-DOM; MFN-has-GEN-has-MFN; TIS-dwr+upr-GEN-dwr+upr-TIS; or PWY-ass-GEN-ass-PWY. GENs from the PGN-pup-GEN or PGN-pdw-GEN are linked through heterogeneous or directed homogeneous associations but not through undirected homogeneous associations.

**Length > 3 (>L3)**. Generated when mapping the source or target PGN to the perturbed genes in L3 metapaths.

In the case of directed homogeneous associations, we used the '_' mark next to the entity that acted as the source of the association. For instance, GEN-_pho-GEN-ass-PWY links the kinases to the pathways associated with their substrates while GEN-pho_-GEN-ass-PWY links the substrates with the pathways associated with their kinases.

Finally, metapaths whose embedding did not preserve the original network or that failed to keep most of the nodes in a single connected component were removed as described in the following section. The entire list of the embedded metapaths is provided in Supplementary Data 3.

## Obtaining Bioteque embeddings

To obtain the embeddings we used the node2vec algorithm[31], a well-accepted approach based on random walk trajectories[78], in which metapaths are used as single networks and fed to the node2vec algorithm. We acknowledge that there are embedding methods that allow a direct embedding of the network from metapath walks (e.g., metapath2vec[79]). However, we decided to first pre-compute the source-target networks using the DWPC method, since the resulting network already weighs those source-target associations that are more strongly connected according to the metapath, thus requiring fewer random walker steps to learn the relationship between the source and target nodes. Moreover, this pre-computed network encourages the embedding model to only focus on source-target relations, giving us more control about what information we are encoding in the embedding space while allowing an easier generalization of the model's hyperparameters across different metapaths lengths (i.e., the source and target nodes are always one-hop apart regardless of the metapath length). Notice that, since all our metapath networks are either homogeneous or bipartite, the default skip-gram implementation of metapath2vec is equivalent to node2vec.

**Homogeneous and bipartite networks**. L1 metapaths already correspond to homogenous or bipartite networks. For >L1 metapaths, the source and target nodes were connected by computing degree-weighted path counts (DWPC)[11] through the corresponding datasets and associations in the metapath. To this end, we sorted the datasets according to the associations of the metapath, represented them as adjacency matrices and kept the same source (rows) and target (columns) node universe as the target and source nodes of the previous and following datasets, respectively. Following the DWPC method, we first downweighted the degree of the nodes in each of the datasets by raising the degrees to the −0.5 power. We then calculated the DWPC values by concatenating the matrix multiplication from the source to the target dataset. As a result, we obtained a new $n \times m$ matrix where $n$ are the source nodes of the first dataset and $m$ are the target nodes of the last dataset. The values of the matrix are the DWPC between the source and target nodes, which are used as weights during the random walker exploration. Finally, we limited the number of edges for each node to 5% of the total possible neighbours (with a minimum of 3 and maximum of 250 edges per node).

Occasionally, we used more than one dataset within the same association or we combined two metapaths into one. This is a common case for >L1 metapaths with transcriptomic signatures where the two directions (CLL-upr-GEN and CLL-dwr-GEN) are often combined (CLL-dwr+upr-GEN-dwr+upr-CPD). To handle these cases, we first obtained an individual network for each metapath or dataset following the approach detailed above. We then merged all the networks by taking the union of the edges (L1 metapaths) or adding the DWPC values (>L1 metapaths).

At the end of the process, we removed network components that cover less than 5% of the entities from the network. And we also removed from the source metapaths that fail to retain 50% of the total nodes within their network components.

**Node2vec parameters**. The node2vec algorithm consists of a random walk-driven exploration of the network followed by a feature vector learning through a skip-gram neural network architecture.

We implemented a custom random walker (with the node2vec parameters $p$ and $q$ set to 1) and ran 100 walks of length 100 for each node of the network. For >L1 metapaths, we scaled the DWPC values for each node to sum 1 and used them as probabilities to bias the random walker. We used the C++ skip-gram implementation provided by Dong et al.[79] with default parameters to obtain a 128-dimensional vector for each node.

## Accounting for node degree biases

The uneven distribution of information across the different knowledge domains and data sources incorporated in our KG inevitably leads to an uneven number of associations across entities, introducing a bias towards nodes with higher degrees. We implemented several measures to mitigate these biases, not only during the generation of the embeddings, but also in the way distances are calculated.

**Before generating the embedding**. To control the degree of the metapath networks, we implemented the DWPC method (as described in the previous section), which was specifically developed to account for degree biases. Furthermore, we also limited the number of connections a given node can have at the end of the metapath to 5% of the total possible neighbours (with a minimum of 3 and maximum of 250 edges per node). This was implemented since we observed that nodes in longer metapaths often find at least one spurious path to connect to every other node in the network. Although most of them end up having very low weights, the resulting network is very dense, requiring a much larger number of random-walks for the skip-gram model to learn the weight distribution of the network. All these cutoffs were chosen based on the thought exploration made by Himmelstein et al. and after

optimizing for different metapaths in our resource. Importantly, the effect of controlling the degree of the network was fundamental for having embedding spaces of good quality, especially for longer metapaths where these biases get exacerbated due to the combination of high-degree nodes from different datasets (Supplementary Fig. 7).

Additionally, we removed from the KG those nodes whose meaning was too general according to the information content provided in the ontology. This prevented those nodes to attract many connections in the network at the cost of providing very little information (e.g., disease terms such as 'cancer', 'syndrome' or 'genetic disease'; or cell compartments terms such as 'cell', 'membrane' or 'cell periphery'). All the pruned terms are provided in Supplementary Data 1.

**After generating the embedding.** Most downstream analyses rely on distances between the embeddings. However, even if we have implemented measures to control the degree of the network when producing the embedding, it is expected that nodes having more general implications will be generally closer to the rest than others that are more specific (e.g.'Brain disease' (https://disease-ontology.org/term/DOID:936) will be closer to a much broad set of genes than 'Migraine' (https://disease-ontology.org/term/DOID:6364) which is a specific condition comprised within the family of Brain diseases). Therefore, some terms may be biased to have a closer distance distribution than others just because their edges define broader associations. Although encoding this can be useful in some downstream analysis (e.g., identifying drugs that target proteins specifically associated with particular brain diseases) it also may introduce biases when comparing distance distributions between terms (Supplementary Fig. 7).

To address these biases, we first assessed how different distances differentiate between these terms, finding that cosine distances provided more comparable distributions between terms while still preserving the (expected) enrichment of small distance associations of broader terms. Moreover, in order to add a measure of specificity in the distance, we also opted to compute co-ranks quantiles, which requires both nodes to be close to each other in order to consider they are sharing a close relationship (this was used in the HuRI-III exercise and the procedure is detailed in the corresponding section). By doing that, we can normalize the distance values of all entities, making them comparable (e.g., having a 0.1 co-rank quantile means the same regardless of the disease node).

Additionally, network permutations can be used in downstream analysis to control spurious observations made in networks that are being analysed with our embeddings. In fact, in the HuRI-III analysis, we randomly permuted the HuRI-III network (as detailed in the corresponding section) and used the permuted network as a reference to derive statistical significance cutoffs for the embedding distances we calculated.

## Embedding evaluation

We used opt-SNE to generate the 2D representation of the embeddings[80]. To assess the quality of the embeddings, we reassembled the network obtained from the metapath using the embedding vectors. To this end, we first computed the cosine distance of each edge in the network using the embedding vectors of the nodes. Next, we generated 100 random permutations for each edge in the network and calculated the cosine distances between them. Finally, we sorted all the distances and computed the area under the ROC curve (AUROC) using the network edges and the random permutations as the positive and negative sets, respectively. When assessing >L1 metapaths, we repeated the same exercise using 3 extra network subsets obtained by keeping, for each node, the top 1%, 25% and 50% closest neighbours according to the DWPC weights of their edges. Embeddings with an AUROC below 0.8 were removed from the resource.

## Embedding characterization

To characterize the embeddings, we first preselected a collection of reference networks representing commonly used biological associations. Then, given a set of embeddings corresponding to a certain metapath, we tested their capacity to recapitulate edges from other (orthogonal) datasets (i.e., the reference networks). Two measures were kept, the coverage (i.e., the number of overlapping nodes) and the AUROC, following the approach described above.

Aiming to extend this characterization, for each metapath we sought to characterize nodes separately, based on their entity type. We first calculated the term frequency-inverse document frequency (TF-IDF) values of the nodes from each reference network in our collection. Next, within the same entity type and network, we used the TF-IDF-transformed vectors to compute pairwise cosine similarities between nodes. Finally, we built the entity similarity network by keeping the top 5 closest neighbours for each node. Note that from one heterogeneous (bipartite) network this process yields two homogeneous networks, one for each entity type.

Some of the networks in our collection required customized preprocessing. To represent perturbation associations, we directly linked the perturbed genes (PGN-pup-GEN or PGN-pdw-GEN) and the outcome of such perturbation (e.g., PGN-bfn-CLL or PGN-upr-GEN) through the corresponding associations and datasets. We computed the CHE-has-CPD similarity networks by directly linking each node with the top 3 partners that shared more neighbours. Additionally, some entity similarity networks were gathered from other sources, like the CPD-CPD mechanism of action similarity obtained from our Chemical Checker resource[81].

## Embedding-based gene expression analysis of cancer cell lines

We downloaded the RMA-normalized gene expression (GEx) and the drug sensitivity data from the GDSC1000[40] web resource (https://www.cancerrxgene.org). We mapped the cell lines and genes to our reference vocabularies and took the mean value whenever duplicates occurred. We used the tissue of origin annotations from the CLUE cell app (https://clue.io/cell-app), which were already part of our graph (CLL-has-TIS, cl_tissue_clueio). Regarding CCLE data, we used the next-generation data[43] from the Broad Institute Portal (https://portals.broadinstitute.org/ccle/about). We processed the RNAseq data and produced three embeddings (CLL-upr-GEN, CLL-dwr-GEN and CLL-dwr+upr-GEN-dwr+upr-CLL) following the pipeline detailed in the "Dataset standardization" and "Obtaining the embeddings" sections.

In the drug sensitivity prediction exercise, we trained a random forest (RF) classifier for each drug and each GEx input data (i.e., the raw GEx or any of the GEx-derived embeddings). After removing drugs with less than 10 sensitive or resistant cell lines, we modelled 262 drugs. We used the SciKit Learn implementation of the RF algorithm, with a 10-fold stratified cross-validation scheme, and optimized RF hyperparameters over 20 iterations of Hyperopt[82].

## Analysis of the HuRI-III protein-protein interaction network

We downloaded HuRI-III from the Human interactome atlas (http://www.interactome-atlas.org/). Next, we considered all L1 metapaths containing a GEN metanode, keeping the dataset with higher coverage for each metapath and discarding those covering less than 10% of the HuRI-III network. As a representative of PPI interactions (GEN-ppi-GEN), we used a version of IntAct dated December 2019 (before publication of the HuRI-III network) from which we removed all entries belonging to the HuRI-III screening (IMEX: IM-25472). Next, we calculated the cosine distance between each PPI in each of the metapath embedding spaces and ranked the distances according to the distance distribution of each of the proteins. Distances and rankings were obtained with FAISS[83]. To derive empirical P values, we transformed the rankings into percentiles by normalizing them by the total number of

covered genes in each metapath and kept the geometric mean of the normalized co-ranked pairs.

In parallel, we generated 1000 random permutations of HuRI-III by randomly swapping each of the HuRI-III edges 10 times using the BiRewire bioconductor package (https://doi.org/10.18129/B9.bioc.BiRewire) and, likewise, calculated P values for each metapath. For each permuted network, we calculated the recovery of the edges with a sliding P value cutoff (between 1 and 0.001) and averaged the counts at each cutoff. After repeating this process with the HuRI-III network, we were able to derive, for each metapath, the expected fold change (FC) across different P value cutoffs (i.e., the number of covered HuRI-III edges at a given P value cutoff divided by the average number of covered edges in the permuted networks). Moreover, the permuted networks were also used to estimate an empirical FDR for a given P value. For instance, for each metapath, we found the P value cutoff associated with a 0.05 FDR by calculating the minimum P value needed to cover no more than 5% of the permuted network edges. Finally, to build the matrix shown in Fig. 6a, we selected the top 20 metapaths with the highest FC (i.e., FC average in the P value range between 0.1 and 0.001), and used their P values to cluster the PPIs with the fastcluster package[84] and the ward distance update formula.

To obtain the Shapley values, we trained a XGBoost model to classify GEN–GEN edges as positive (i.e., present in HuRI-III) or negative (i.e., not present in HuRI-III) using the P values across metapaths as features. To sample negative pairs, we used the instance of the permuted networks hitting fewer HuRI-III edges (~3%) in order to avoid having the same edge as positive and negative instance at the same time. Furthermore, since the objective of this exercise was to study the interplay between the metapaths, we removed edges that were covered by less than 10 (50%) metapaths, resulting in a dataset of 60k positive and negative pairs. A simple mean imputation was applied to the missing P values. At training time, we implemented a 20-fold stratified cross-validation split scheme and fine-tuned the hyperparameters using 20 iterations of Hyperopt[82]. Finally, we obtained the Shapley values from the test splits by implementing the TreeExplainer method[49]. All subsequent analyses and figures were obtained using the SHAP package (https://github.com/slundberg/shap).

### Drug repurposing based on drug and disease embeddings

The first release of the repoDB (v1) data was downloaded from http://apps.chiragjpgroup.org/repoDB while the updated release (v2) was obtained from https://unmtid-shinyapps.net/shiny/repodb. Compounds were mapped to InChIKeys and diseases to the Disease Ontology (DO) forcing a 1:1 mapping. As features, we used the following metapaths (datasets) from the Bioteque resource: CPD-int-GEN (curated_targets); DIS-ass-GEN (disgenet_curated+disgenet_inferred); CPD-int-GEN-int-CPD-has-PHC (curated_targets-curated_targets-atc_drugs); and DIS-ass-GEN-ass-DIS-trt-CPD (disgenet_curated+disgenet_inferred-disgenet_curated+disgenet_inferred-repodb).

Additionally, we obtained the 2048-bit Morgan fingerprints (ECDF4) of the compounds using RDKIT (http://rdkit.org) and used the adjacency matrix of the disease-gene network from DisGeNET as binary descriptors of diseases. Having defined the features of the model, we filtered out those drugs and diseases from repoDB that fell outside the embedding universe and removed redundant pairs by de-propagating the associations to the most specific drug-disease terms according to the Disease Ontology. As a result, the train (repoDB v1) and test (repoDB v2) splits consisted of 2522 and 1187 unique drug-disease associations, respectively (Supplementary Fig. 5). Additionally, to prevent the model from focusing on the most frequently annotated drug and disease entities, we further processed the train data to

balance the number of associations (degree of the nodes). More specifically, we capped the number of drug or disease associations to 5% of all possible associations (44 diseases and 26 drugs, respectively). Therefore, the associations of those drugs or diseases exceeding this limit were subsampled by performing a K-means clustering (where K was set to the capping limit) using the CPD-int-GEN or DIS-ass-GEN embeddings as features, and by randomly selecting a representative association from each of the clusters (Supplementary Fig. 5). This step slightly decreased the number of training data to 2326 drug-disease associations.

Next, we produced train negative pairs by aggregating 20 negative networks obtained by randomly swapping the edges of the training data (thus, forcing a ratio of 1:20 between the positive and negative instances), while preventing inconsistencies in the Disease Ontology (i.e., having a negative association that would be obtained by propagating a positive drug-disease association through the ontology). Note that, to comply with the time-split scenario, we did not remove any negative drug-disease pair reported to be positive in the repoDB v2 release.

Once the training data was ready, we ran an RF classifier for each of the explored models using 20 iterations of Hyperopt[82] to fine-tune the hyperparameters. At prediction time, drug-disease associations in repoDB v2 were considered positive test pairs, whereas all the remaining drug-disease pairwise combinations were considered negative pairs. To avoid inconsistencies, we removed those negative pairs that were semantically related to positive pairs according to the Disease Ontology. As a result, we obtained between [460–500] diseases and [750–800] drug predictions for each drug and disease, respectively. As most of the drugs and diseases only had one or two positive instances, we assessed the performance of the models by ranking all the predictions individually for each entity (ranks were used as percentages). Additionally, we calculated ROC curves for those drugs and diseases that had at least 5 positive instances. Finally, we obtained the pharmacological action of the drugs by mapping them to the uppermost level of the Anatomical Therapeutic Chemical (ATC) classification, when available. Likewise, disease families were derived by propagating the disease terms to the first and second levels of the Disease Ontology.

### Reporting summary

Further information on research design is available in the Nature Research Reporting Summary linked to this article.

## Data availability

All the embeddings generated in this study are available for direct download from https://bioteque.irbbarcelona.org/downloads. The raw networks that were embedded are provided in the same downloadable file for metapaths of length ≥ 2. To comply with the wide variety of licences associated with the data owners, raw networks for L1 metapaths are not provided. Instead, instructions and code to download and pre-process the data are made available at https://gitlabsbnb.irbbarcelona.org/bioteque/. Accessible links to all the datasets embedded in the Bioteque resource are listed on https://bioteque.irbbarcelona.org/sources. RMA-normalized expression data of the GDSC cell lines was downloaded from https://www.cancerrxgene.org/gdsc1000/GDSC1000_WebResources/Home.html. CCLE RNAseq data was downloaded from https://sites.broadinstitute.org/ccle/datasets. Cell line tissue of origin annotations were obtained from clue.io (https://clue.io/cell-app). The HuRI-III network was downloaded from http://www.interactome-atlas.org/download. The first release (v1) of repoDB indications was downloaded from http://apps.chiragjpgroup.org/repoDB/. The second release (v2) of repoDB indications was downloaded from https://unmtid-shinyapps.net/shiny/repodb. ATC codes were obtained from Drugbank (https://go.drugbank.com/

releases/latest#full), Drugcentral (https://drugcentral.org/download) and KEGG (https://www.genome.jp/kegg-bin/get_htext?br08303+ D00731). Curated gene-disease associations were downloaded from DisGeNET (https://www.disgenet.org/downloads).

## Code availability

The code used to generate the embedding resource is available at https://gitlabsbnb.irbbarcelona.org/bioteque/. Individual scripts used to download, pre-process and integrate the embedded datasets into the knowledge graph can be obtained from https://bioteque.irbbarcelona.org/sources. Jupyter notebooks exemplifying how to programmatically download embeddings from the Bioteque resource and how to run the downstream tasks illustrated in this manuscript can be downloaded from https://bioteque.irbbarcelona.org/downloads/demo.

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

## Acknowledgements

P.A. acknowledges the support of the Generalitat de Catalunya (RIS3-CAT Emergents CECH: 001-P-001682 and VEIS: 001-P-001647), the Spanish Ministerio de Ciencia, Innovación y Universidades (PID2020-119535RB-I00), the Instituto de Salud Carlos III (IMPaCT-Data), and the European Commission (RiPCoN: 101003633). A.F-.T. is a recipient of an FPI fellowship (BES-2017-083053). We also acknowledge institutional funding from the Spanish Ministry of Science and Innovation through the Centres of Excellence Severo Ochoa Award, and from the CERCA Programme/Generalitat de Catalunya.

## Author contributions

A.F.-T., M.D.-F. and P.A. designed the study and wrote the manuscript. A.F.-T. implemented the entire computational strategy and analysis. A.F.-T., M.B. and M.L. implemented the web resource. All authors analyzed the results and read and approved the manuscript.

## Competing interests

The authors declare no competing interests.
