## [Peer Review File · Nature Communications]

Reviewers' Comments:

Reviewer #1:

Remarks to the Author:

The authors present 'the Bioteque': an impressively large repository of pre-calculated biomedical knowledge graph embeddings. Bioteque builds on a manually curated knowledge base hypergraph linking together multiple biological entities (for example genes, compounds, cell lines, tissues, diseases etc.) and the relation occurring among them (such as for example a gene being up/down-regulated in a disease state versus normal state, or being targeted by a compound, or a cell line being derived from a given tissue etc.).

The authors have implemented an established (and previously published) procedure that starting from this knowledge based hypergraphs builds 'graph embeddings' between n-ples of individual node classes, i.e. bidimensional visualisations in which points are nodes of the original graph and their proximity reflects topological similarities (in terms of shared random walk paths) in the original graph when considering a limited number of node classes (which are first determined to build methapaths, for example between compounds - genes - diseases).

The authors show that this strategy produces euclidean embeddings with minimal introduced distortions and that overall preserve the original KG topology, i.e. with proximal nodes more likely to be directly connected in the original KG.

Furthermore the author show that embeddings of different types can be analysed to infer relations between nodes (for example classifying cell lines nodes based on their tissue of origin, passing through up/down-regulated genes layer, or predicting drug repurposing opportunities or cell line responses to individual drug treatments).

Briefly, this manuscript presents an important resource, that for the first time integrates a large number of heterogenous datasets and produces an excellent means to analyse them jointly, and will arguably support in the future queries that are formulated in natural language. The resource is clearly presented and each construction and assessment step well documented.

I only have few minor points, which if addressed might contribute to further improve this (already publishable) manuscript:

- As all the embeddings are visualised in a two dimensional euclidean plane wouldn't make more sense to assess that they retain the original KG interactions using euclidean distance rather than cosine distance?
- The authors refer to 'embeddings' and other jargon throughout the text, right from the abstract, but specifying the meaning of most very graph-theory specific terms. This should be avoided, and the concept of 'embedding' for example should be defined at its first instance for readers that are not familiar with it.
- I find most figure panels difficult to follow due to too many abbreviations, i.e. one for each node and each edge type, The authors might consider removing some of these (at least for the node classes or edge types with short names or with names composed of a single word, such as genes, cells, diseases, has, interacts, similar, ...)
- It is not clear how different node degrees (reflecting an uneven distribution of prior knowledge across the bioteque due to many biological entities being inherently more characterised than others, just due to the fact that the have been studied for longer, for example) is accounted by the authors while computing their metapaths necessary to the embedding calculations. The author mention that they down-weight highly connected nodes, but this deserves a better explanation/justification to me. The authors for example might consider to incorporate in their tool a network randomisation procedure that preserves nodes degrees and could be used to 'normalise' outputted metapaths. In this respect, the tools presented in PMID: 25161255 and in PMID: 27998275 could be used.

Reviewer #2:

Remarks to the Author:

This manuscript describes a new resource, the Bioteque, that provides access to the set of embeddings describing a knowledge graph that integrates a range of systematic data linking genes, diseases, drugs, gene expression data and so on. The use of a knowledge graph for this kind of data is not unique. There are several examples out there in this domain including Paliwal, S. et al (<https://doi.org/10.1038/s41598-020-74922-z>), Geleta, D. et al. (<http://biorxiv.org/lookup/doi/10.1101/2021.10.28.466262>), Santos, A. et al. (doi:10.1038/s41587-021-01145-6), Sosa, D. N. et al. (Pac. Symp. Biocomput. Pac. Symp. Biocomput. 25, 463–474) and so on. The difference with this resource is that the embeddings are provided through the public resource, <https://bioteque.irbbarcelona.org>, to enable others to use them to construct their own analyses. In general this is a great innovation which I applaud.

The manuscript could be improved in several ways.

First, if possible there should be a wider comparison with the existing knowledge graph implementations in terms of their design and data coverage. This can be problematical as some graphs are not available, but it is possible to summarise differences in design choice. As I understand the methods for some datasets with continuous scores, the data is converted into a binary Y/N edge. How does this affect the graph and the embeddings? Could weighted edges be used instead?

Second, as well as the summary explorations of the applications of the embedding approaches, there should be some deeper analysis of the value of these approaches. For instance what concrete examples are there of the drug repurposing analysis? Would a drug discovery scientist think that these make sense? If the prediction is conducted using historical data, can new repurposing predictions be validated by later entry into clinical trials? While individual examples can be cherry picked, and of course the global analysis is required, they also shed light on the quality of the predictions. Again for example, the use of drugs with TOP2A as target in Leukemia and Kaposi's Sarcoma is picked out as being a novel relationship between these diseases. However it's not clear what the novelty is because these drugs are widely prescribed for diseases involving rapid cell proliferation which includes both indications.

Third, given that the novelty of the approach is the embeddings and the ability to use these for future analyses, it would be helpful to have examples of how this could be done. How does one include a new data set in this system? What steps does the user need to take to do this?

Minor corrections

- Fig 2C – misspelling of length
- OpenTargets should be Open Targets

Reviewer #3:

Remarks to the Author:

Review -- Integrating and formatting biomedical data in the Bioteque, a comprehensive repository of pre-calculated knowledge graph embeddings.

In this manuscript the authors describe a resource, the Bioteque, that provides node embeddings pre-calculated from a comprehensive biomedical graph representing relationships between multiple biological entities. The authors present several examples where these embeddings or vector representations are shown to retain biological information and can be used in machine learning tasks of different purpose. This shows a great effort of data integration built upon previous initiatives such as the Harmonizome or Hetionet and generates a resource that opens multiple possibilities although the usability of the resource could be improved.

Embeddings

- The authors generate embeddings using the Node2Vec algorithm and avoid its limitation of not being suitable for heterogeneous networks by representing homogeneous graphs or applying DWPC. Algorithms such as Metapath2Vec (<https://doi.org/10.1145/3097983.3098036>) can handle representation of heterogeneous graphs generating random walks over heterogeneous metapaths and incorporate them into skip-gram model. Could the authors describe why their approach would be preferred?
- The authors mention that the aim of the paper is not to benchmark different embedding algorithms and they cite the Node2Vec paper where different shallow embedding methods are compared. However, it would be relevant to compare results from two different learning techniques: shallow vs deep. Algorithms such as Graph Neural Networks can account for more complex network structures than the ones used (heterogeneous, node features) and they may provide more meaningful embeddings and help to represent unseen entities, which may be useful when evaluating a new dataset, compound, etc. These algorithms and their benefits and drawbacks should be at least discussed in the manuscript. I believe the comments in the discussion regarding “improvement of embedding strategies” points in this direction, but I would suggest making this more explicitly and openly discuss what the limitations of the current approach and how it could be improved.

Data integration

- Some of the integrated resources provide diverse scores to weight the associations, i.e STRING, DisGENET, COMPARTMENTS, etc. In the manuscript is not mentioned whether the authors used this information and how. If they did not use these scores, it would be relevant to evaluate how low-confidence scores influence the structure of the graph and thus the embeddings generated.
- Looking at the resources ingested, there are in some cases associations integrated from multiple resources. The authors looked at overlap for instance in GEN-*assoc*-Disease pairs (Figure 1E), but the comparison of up- and down-regulated genes would be very relevant. Did the authors look at the consistency across expression sources? It would be important to evaluate the quality of the integrated datasets to avoid including inconsistencies that could lead to noise influencing the embeddings.
- In Figure 1F, it seems that the most popular genes, compounds, and pathways point toward cancer, however, only few cancer types show up as popular. This is also surprising given the focus on this disease of some of the resources included (Achilles project, COSMIC, etc.). Could the authors comment on this?

Usability

- The integration effort is one of the most valuable pieces of this work, however, the graph built is not provided to the community. This graph would be very valuable firstly to evaluate the quality of the graph and secondly to use it for further exploration and investigation. The authors could provide the graph in a standard format or tabular edge lists.
- The web interface is not sufficiently promoted in the manuscript, and it is an important part of this work. The plots in the web provide the same information as in some of the figures of the manuscript, however, they provide very little interaction and very little information. For instance, the 2D projection provides very little information about which entities are close to each other or what are the clusters. It would be very useful to be able to identify the dots in the 2D projection by either hovering or searching for specific entities. Further, other annotations such as tissue information, compound class, etc. could be used to colour genes or compounds to show whether clusters represent biological information.
- I congratulate the authors for providing a link to the repository where the pre-processing scripts can be found! The documentation regarding how these scripts can be used is very limited and it would still make it difficult to replicate the results obtained. I would recommend the authors to provide enough documentation to be able to run and test the scripts provided making use at least of the README files already included to provide examples.

Resource update

- The authors mention that the resource and embeddings will be updated yearly or as soon as there are major new versions of datasets. This is excellent, but it would be good if the authors could detail how new versions will be checked for quality (i.e “how much” embeddings changed, how predictions deviated, etc) and how they will report these metrics. This would be important to ensure the quality of the resource and the validity of previously used embeddings. This is linked to

the previous point about the quality of the data integrated, which has an impact on the representations generated.

Point-by-point responses to Reviewer's comments

NCOMMS-22-17971 "Integrating and formatting biomedical data in the Bioteque, a comprehensive repository of pre-calculated knowledge graph embeddings" by Fernández-Torras, et al.

We would like to thank the Editor and Reviewers for their critical reading of our manuscript and the positive feedback. We have now addressed in detail all the raised issues, which, we feel, has improved the quality and clarity of our paper. In brief:

- We have extended the Methods section to add further clarifications and justifications of the experimental choices we have made for creating our resource.
- We have improved both our web resource and GitHub pages, vastly extended the documentation to detail how we process the raw data from the different sources to generate the embeddings, and prepared jupyter notebooks to illustrate how to download and perform most of the downstream analyses presented in the m/s.
- We have added a new **Supplementary Table 2**, two new **Supplementary Figs. (1 and 8)**, and extended **Supplementary Fig. 7** and **Supplementary Tables 3** and **5** to support the new analyses of the results in response to the Reviewers comments. Please, notice that supplementary figures and tables have been renumbered to accommodate the new additions.
- We have improved the clarity of the text and added graphical support legends to the main figures.

Please find below a point-by-point response to Reviewers' comments. Major edits are highlighted in yellow in the revised version of the manuscript.

Reviewer #1 (Remarks to the Author):

The authors present 'the Bioteque': an impressively large repository of pre-calculated biomedical knowledge graph embeddings. Bioteque builds on a manually curated knowledge base hypergraph linking together multiple biological entities (for example genes, compounds, cell lines, tissues, diseases etc.) and the relation occurring among them (such as for example a gene being up/down-regulated in a disease state versus normal state, or being targeted by a compound, or a cell line being derived from a given tissue etc.).

The authors have implemented an established (and previously published) procedure that starting from this knowledge based hypergraphs builds 'graph embeddings' between n-ples of individual node classes, i.e. bidimensional visualisations in which points are nodes of the original graph and their proximity reflects topological similarities (in terms of shared random walk paths) in the original graph when considering a limited number of node classes (which are first determined to build methapaths, for example between compounds - genes - diseases).

The authors show that this strategy produces euclidean embeddings with minimal introduced distortions and that overall preserve the original KG topology, i.e. with proximal nodes more likely to be directly connected in the original KG.

Furthermore the author show that embeddings of different types can be analysed to infer relations between nodes (for example classifying cell lines nodes based on their tissue of origin, passing through up/down-regulated genes layer, or predicting drug repurposing opportunities or cell line responses to individual drug treatments).

Briefly, this manuscript presents an important resource, that for the first time integrates a large number of heterogenous datasets and produces an excellent means to analyse them jointly, and will arguably support in the future queries that are formulated in natural language. The resource is clearly presented and each construction and assessment step well documented.

I only have few minor points, which if addressed might contribute to further improve this (already publishable) manuscript:

We would like to thank the Reviewer for his/her very positive comments on the novelty and the value of our resource. We have now addressed the highlighted points, which has indeed improved the clarity and robustness of the m/s.

As all the embeddings are visualised in a two dimensional euclidean plane wouldn't make more sense to assess that they retain the original KG interactions using euclidean distance rather than cosine distance?

The selection of the metrics used to calculate distances between embedding vectors is, indeed, an important issue, since different distances may lead to different performances depending on the exercise or the methodology followed to obtain the embedding space. First, we want to highlight that all the distances computed in this manuscript are based on the 128 dimensions of the vectors, and not on the 2D projections, so that we use all the information encoded by our embedding model. Having said that, we actually use Euclidean distances as the underlying metric for the t-SNE to compute the 2D representations. Regarding the assessment of how the embedding space retains the original KG interactions and other orthogonal associations, our pipeline computes an AUROC score using both Euclidean and cosine distances. However, we observed that cosine distances usually perform better in both exercises (**Fig. R1**), so we decided to keep cosine distance as the reference distance. This is in good agreement with other studies that observed that cosine distances are better suited to recapitulate word semantic similarities between word embeddings (e.g. [arXiv:1510.02675](https://arxiv.org/abs/1510.02675) or [arXiv:1508.02297](https://arxiv.org/abs/1508.02297)). Besides, we also observed that cosine distances provide more comparable numerical ranges between entity types than Euclidean distances do, as we illustrate in our reply to the last point asked by the Reviewer (**Fig. R3**).

Figure R1. Comparison between cosine and Euclidean distances. **Left)** We provide the AUROCs score obtained when assessing the retention of the original KG interactions for each metapath-dataset in our resource according to the cosine (y-axis) and Euclidean (x-axis) distances. Notice that, while both distances are highly correlated (Pearson: 0.81; Spearman: 0.87), cosine distances tend to be better at recapitulating the original network. **Right)** The AUROCs scores when recapitulating orthogonal KG associations were first computed using either Euclidean or cosine distances. We then subtracted the Euclidean-based AUROC score from the cosine-based score (x-axis) and computed the cumulative distribution across all the metapath and datasets in our resource (y-axis). To avoid considering differences within uninformative ranges, we removed those measures where both distances had an AUROC below 0.6. This analysis reveals that over 80% of all the scores have a better AUROC when computations are based on cosine distances. Significantly, 25% of all the associations have a 0.05 gain when using cosine distance, while only 5% of them achieve this gain when using Euclidean distances.

To accommodate Reviewer's suggestion, we have modified the manuscript to include the network preservation score (i.e., retention of original KG interactions) based on Euclidean distances (**Supplementary Table 5**).

The authors refer to 'embeddings' and other jargon throughout the text, right from the abstract, but specifying the meaning of most very graph-theory specific terms. This should be avoided, and the concept of 'embedding' for example should be defined at its first instance for readers that are not familiar with it.

We agree with the Reviewer that some terms widely used in the manuscript may be too specific for a particular field, and we thank the Reviewer for noticing this. Following the Reviewer's suggestion, we have thoroughly revised our manuscript to now provide a definition for those specific terms when they are first mentioned in the text. Additionally, the manuscript has now been seen by our Editorial Support department, which has also checked the clarity of the wording for a non-specialized audience.

I find most figure panels difficult to follow due to too many abbreviations, i.e. one for each node and each edge type. The authors might consider removing some of these (at least for the node

classes or edge types with short names or with names composed of a single word, such as genes, cells, diseases, has, interacts, similar, ...)

We agree with the Reviewer that the use of abbreviations instead of full labels makes the figure panels more difficult to follow. However, the use of abbreviations representing the biological entities and their relationships is a fundamental part of our resource, being used not only in the metapath nomenclature but also to annotate the recapitulation of orthogonal associations in the KG. Therefore, for the sake of consistency and to help the reader familiarize themselves with this nomenclature, we think that these abbreviations should be kept as much as possible. Nevertheless, to provide a smoother read of the figure panels as pointed out by the Reviewer, we included a legend in each figure where abbreviations are mapped to their full name.

It is not clear how different node degrees (reflecting an uneven distribution of prior knowledge across the bioteque due to many biological entities being inherently more characterised than others, just due to the fact that they have been studied for longer, for example) is accounted by the authors while computing their metapaths necessary to the embedding calculations. The authors mention that they down-weight highly connected nodes, but this deserves a better explanation/justification to me. The authors for example might consider to incorporate in their tool a network randomisation procedure that preserves nodes degrees and could be used to 'normalise' outputted metapaths. In this respect, the tools presented in PMID: 25161255 and in PMID: 27998275 could be used.

The Reviewer raises a very important point. Indeed, the distribution of knowledge is uneven between entities, which ultimately affects their degree in the network and introduces undesired biases. For this reason, we implemented different measures to control these biases, not only during the generation of the embeddings but also in the way distances are calculated as well as how we addressed some of our analyses.

Before generating the embedding

To control the degree of the metapath networks (before generating the embedding), we implemented the DWPC method described by Himmelstein et al. (PMID: 26158728), which was specifically developed to address this problem when dealing with metapaths. Following this method, we individually down-weight each path between a source and target node by first raising the degree of each node to the -0.5 power and then multiplying all exponentiated degrees of the path. Additionally, we limit the number of connections a given node can have at the end of the metapath. More specifically, we prevent nodes from having more than 5% of all possible network connections or more than 250 interactions. When this happens, we keep the top 5% or 250 edges with higher DWPC weight. This was implemented since we observed that nodes in longer metapaths often find at least one spurious path to connect to every other node in the network. Although most of them end up having very low weights, the resulting network is very dense, making the skip-gram model need many random-walk steps iterations in order to learn the weight distribution of the network. All these cutoffs were chosen based on the thorough exploration made by Himmelstein et. al. and after optimizing for different metapaths in our resource. Furthermore, the 250 edges limit is also convenient as the same was used to keep the top up- and down-regulated genes from cell gene expression datasets. As the Reviewer anticipated, the effect of

controlling the degree of the network is fundamental for having embedding spaces of good quality, especially for longer metapaths where these biases get exacerbated due to the combination of high-degree nodes from different datasets (**Fig. R2**).

Additionally, we removed from the KG those nodes whose ‘meaning’ was considered to be too general according to the information content provided in the ontology. This prevented those nodes to attract many connections in the network at the cost of providing very little information (e.g. disease terms such as ‘cancer’, ‘syndrome’ or ‘genetic disease’; or cell compartments terms such as ‘cell’, ‘membrane’ or ‘cell periphery’). All the pruned terms are provided in **Supplementary Table 4**.

Figure R2. Assessing the effect of DWPC and pruning in the embeddings. Using the Bioteque metapath embedding ‘CPD-trt-DIS-ass-GEN-ass-DIS’ as reference we calculated 3 other embedding spaces where we removed the DWPC weights (w/o DWPC), the limitation in the number of edges (w/o pruning) or both (w/o DWPC, w/o pruning). Looking at the 2D representations (first row) we can see how removing the DWPC or the pruning introduces biases in the space according to the node type, making them cluster separately in the 2D projection. More importantly, this affects the ability of the space to recapitulate the original KG (second row). In the last row, we show the association between the average z-score cosine distance of each

node (y-axis) and their normalized degree (i.e. divided by the max degree within each node type) in the network (x-axis). Notice that, while it is expected that nodes with a higher degree will be, by definition, closer to more nodes, the average distance does not differ more than 1 standard deviation from the average (z-scores between -1 and 1). However, removing either the DWPC or the pruning makes higher-degree nodes much closer, on average, to any other node in the space. Consequently, these biases the tail of the distribution representing small distances towards those associations involving high-degree nodes.

After generating the embedding

Most downstream analyses rely on distances between the embeddings. However, even if we have implemented measures to control the degree of the network when producing the embedding, it is expected that nodes having more general implications will be generally closer to the rest than others that are more specific (e.g. 'Brain disease' (DOID:936) will be closer to a much broad set of genes than 'Migraine' (DOID:6364) which is a specific condition comprised within the family of Brain diseases). Therefore, some terms may be biased to have a closer distance distribution than others simply because their edges define broader associations. Although encoding this can be useful in some downstream analyses (e.g. identifying drugs that target proteins specifically associated with particular brain diseases), it may also introduce biases when comparing distance distributions between terms (**Fig. R3**).

Figure R3. Distance distribution of the 'Brain disease' and 'Migraine' nodes to each of the genes available in the GEN-ass-DIS embedding space (obtained from DisGeNET). From left to right we show the distribution using Euclidean distances, cosine distances (1-cosine similarity), and Co-rank quantiles (calculated as specified in the Methods section).

To address these biases, we first assessed how different distances differentiate between these terms, finding that cosine distances provided more comparable distributions between terms while still preserving the (expected) enrichment of small distance associations of broader terms. This observation may partially explain why cosine distances are better at recapitulating the KG network than Euclidean distances (as shown in **Fig. R1**). However, in order to ensure that distance measures are relatively specific, we also opted to compute co-ranks quantiles, which requires both nodes to be close to each other (this approach was used in the HuRI-III exercise). By doing

that, we can normalize the distance values of all entities, making them comparable (e.g having a 0.1 co-rank quantile means the same regardless of the disease node).

On the other hand, we acknowledge there are other methods to account for these biases such as the network randomization methods suggested by the Reviewer. We decided to use DWPC since it was specifically designed and optimized for dealing with metapaths (being significantly fast to compute) and the authors showed it works better than other weight-based alternatives (PMID: 26158728). Having said that, network permutations can be used in downstream analyses to control spurious observations made in networks that are being analyzed by the embeddings of our resource. In fact, in the HuRI-III analysis, we randomly permuted the HuRI-III network using BiRewire (PMID: 27998275; i.e. one of the methods suggested by the Reviewer), and used the permuted network as a reference to derive statistical significance cutoffs for the embedding distances.

While all this information was provided through the methods section, we agree with the Reviewer that the importance of dealing with high-degree nodes is not explicitly discussed in the manuscript. Thus, following the Reviewer's suggestion, we have extended the methods to introduce a section in which we explicitly detail all the actions we took to address these biases, and included **Fig. R2** and **R3** as **Supplementary Fig. 8**.

Reviewer #2 (Remarks to the Author):

This manuscript describes a new resource, the Bioteque, that provides access to the set of embeddings describing a knowledge graph that integrates a range of systematic data linking genes, diseases, drugs, gene expression data and so on. The use of a knowledge graph for this kind of data is not unique. There are several examples out there in this domain including Paliwal, S. et al (<https://doi.org/10.1038/s41598-020-74922-z>), Geleta, D. et al. (<http://biorxiv.org/lookup/doi/10.1101/2021.10.28.466262>), Santos, A. et al. (doi:10.1038/s41587-021-01145-6), Sosa, D. N. et al. (Pac. Symp. Biocomput. Pac. Symp. Biocomput. 25, 463–474) and so on. The difference with this resource is that the embeddings are provided through the public resource, <https://bioteque.irbbarcelona.org>, to enable others to use them to construct their own analyses. In general this is a great innovation which I applaud.

We thank the Reviewer for his/her kind words about the innovation of our approach, and for noting and expressing the added value of our work with respect to other KGs. Indeed, we have added these citations to the main m/s to help the reader better contextualize it.

The manuscript could be improved in several ways.

First, if possible there should be a wider comparison with the existing knowledge graph implementations in terms of their design and data coverage. This can be problematical as some graphs are not available, but it is possible to summarise differences in design choice.

We agree that a comparison to other KGs would help visualize the data integration effort we made to construct the resource and thank the Reviewer for noticing it. Consequently, to help contextualize our KG with other existing graphs, we have taken the thorough comparison made by Bonnet et al. ([arXiv:2102.10062](https://arxiv.org/abs/2102.10062), Table 15) and included an extra row representing our KG

(Table R1). We refer the Reviewer to the original Bonnet et al. study in case they need further information regarding the other KGs used in this comparison.

KG dataset	Design Use case	Entities	Edges	Entity Types	Relation Types	Contains Features	Datasets	Version info	Last update
Hetionet	Repurp.	47K	2.2M	11	24	no	29	no	2017
DRKG	Repurp.	97K	5.7M	13	107	yes	34	no	2020
BioKG	General	105K	2M	10	17	yes	13	no	2020
PharmKG	Repurp.	7.6K	500K	3	29	yes	7	no	2020
OpenBioLink	Benchmark	184K	4.7M	7	30	no	17	no	2020
Clinical KG	Personalized medicine	16M	220M	35	57	no	35	no	2020
Bioteque	General	450k	30M	12	67	no	150 (66*)	yes**	2022

Table R1. Comparing pre-existing knowledge graphs.

* Although our KG includes up to 150 datasets, we selected 66 as a reference to perform the embeddings.

** We will update the resource yearly and we will provide version information.

In general, our KG is the most comprehensive in the number of processed datasets, the second most comprehensive with respect to entities, edges, and relation types, and the third regarding entity types. However, it is important to highlight that methodological choices followed by these resources have introduced biases in these numbers. For instance, DRKG decided to split disease terms into ‘diseases’, ‘side effects’, and ‘symptoms’ entity types. We, on the contrary, have merged these terms into a single entity type (‘diseases’). Another design choice to highlight is the decision to include ‘publications’ and ‘protein mutational variants’ as entities in the Clinical KG, as only these two cover more than 100M of all the nodes and edges in their graph. Additionally, they also differentiated between ‘protein peptides’, ‘clinical variants’, and ‘protein transcripts’, being them accounting for most of the nodes in their graph (check **Fig. S3** in PMID: 35102292). While increasing the granularity of the entity types can help provide more specific answers for particular queries (e.g. ‘which specific mutational variant is associated with a given disease of interest?’), it also makes the biological connections dilute between different nodes, making the integration of orthogonal data more difficult. For instance, by keeping ‘side effects’, ‘diseases’, and ‘symptoms’ terms in a single entity type (‘diseases’), we are able to obtain a more comprehensive set of biomedical descriptors for the same disease. That is to say, for the same disease node, we can obtain embeddings describing its side effects, symptoms, and gene associations contexts by using the metapaths DIS-cau-CPD, DIS-cau-DIS and DIS-ass-GEN, respectively.

We have now included this comparative table, and the accompanying discussion in the manuscript as **Supplementary Table 2**.

As I understand the methods for some datasets with continuous scores, the data is converted into a binary Y/N edge. How does this affect the graph and the embeddings? Could weighted edges be used instead?

The Reviewer raises a very good point. In fact, when we started building the resource, we considered the addition of weight information to guide the random walker previous to the embedding step. However, we realized that the incorporation of weights to the edges is far from a trivial task. More specifically, most of these weights are based on a measure of support or “confidence”, which may not directly correspond to biological significance/strength. Instead, these scores usually reflect either bias on the knowledge annotation or detectability limitations of the experimental screening (e.g. the expression of some genes may be more difficult to detect than others). For instance, the Open Targets initiative scores disease-gene associations according to the evidence reported in different sources. Although this information can be relevant, it also introduces biases since under-studied diseases will be less covered among the different sources and, therefore, will have lower ‘confidence scores’ by definition. In fact, the same platform warns users to interpret these scores with caution (check the section ‘*Interpreting association scores*’ <https://platform-docs.opentargets.org/associations>). Consequently, the incorporation of these scores as weights would bias the embedding space to keep genes closer to those diseases that are more studied, irrespective of the strength of the biological association of the genes to those particular diseases. Another example is the STRING PPI network, whose scores do not indicate the strength or the specificity of the interaction but its confidence based on accumulated evidence (<https://string-db.org/cgi/info>). Overall, while the incorporation of weighted edges can sometimes include valuable information for the embedding, the peculiarities and possible biases that can be introduced make this step hard to generalize across the diverse and heterogeneous associations in our resource. Therefore, we opted to provide a resource where all the datasets have been treated as evenly as possible.

Having said that, and to address the Reviewer’s comment, we have assessed whether embedding space is able to recapitulate (“predict”) different versions of the network with respect to the score weights of the dataset (**Fig. R4**). Interestingly, this suggests that the associations we are keeping in the network provide enough information to learn the specific context of each node, being this able to recapitulate (without training for it) other interactions that are more likely to happen according to the support provided by the original source.

Figure R4. Network preservation with respect to edge support weights. Taking the STRING (GEN-ppi-GEN) embedding space, we measured the recapitulation of the whole STRING network according to the edge support (weight) provided by the STRING resource. Edge groups that are included in the embedded network (>0.7) are shown with discontinuous lines. Accordingly, straight lines show the recapitulation of edges that were not explicitly embedded and, therefore, can be considered as “link predictions” made by the embedding space. Notice that STRING weights (from 0.7 to 1.0) were not considered when creating the embedding space. Moreover, no model has been trained for this exercise, being this recapitulation obtained by simply calculating distances between the network edges.

On the other hand, in cases where continuous scores do imply a sort of biological strength, we have shown how the embedding space obtained after binarizing the data does preserve the characteristics of the original raw data. This was illustrated in the case of cellular gene expression, where the first binarized and then embedded cell-gene pairs did recapitulate the characteristics observed when using the full vector of continuous gene expression measures (**Fig. 4 A, B and C**). Finally, we want to highlight that, to binarize continuous data, we have followed standard procedures that have been extensively used and validated. For instance, we binarized gene expression data following the Harmonizome (PMID: 27374120) strategy, which is based on conventional statistical treatment of gene expression profiles, while drug sensitivity data have been binarized following the ‘Waterfall method’, first described by Barretina et al. in the publication of the CCLE (PMID: 22460905) and used since then in different works (e.g. PMID: 24284626, PMID: 26570998, PMID: 26656004).

We have added a paragraph to the ‘Dataset Standardization’ method section to explain our choice of binarizing continuous values.

Second, as well as the summary explorations of the applications of the embedding approaches, there should be some deeper analysis of the value of these approaches. For instance what concrete examples are there of the drug repurposing analysis? Would a drug discovery scientist think that these make sense? If the prediction is conducted using historical data, can new repurposing predictions be validated by later entry into clinical trials? While individual examples

can be cherry picked, and of course the global analysis is required, they also shed light on the quality of the predictions.

We agree with the Reviewer that specific examples can help understand the type of findings we are observing from our analysis. Indeed, we make an effort to provide such illustrative examples for those results which we considered to be 'unexpected' (i.e. the observation of compound-disease indications clustering together in an embedding space based on gene-driven associations, since this information was not explicitly used in the metapath). For this reason, we decided to highlight representative compound-disease pairs, and their shared targets, for each of the clusters we identified enriched in this type of association. Individual examples were also provided within the HuRI-III exercise, where we identified PPI pairs that, while not supported by evidence coming from previously published PPI networks, were predicted as putative interactors based on support evidence from other biological knowledge domains (e.g. the interactions between HOMER-SHANK2, TSEN54-CLP1 or ADARB1-PRKRA, **Fig. 5D**). Additionally, we also provide specific examples that illustrate cases in which entity pairs are similar according to a given biological context but different in others (**Supplementary Fig. 3A**).

Regarding the drug repurposing analysis, we would like to highlight that our models were trained on the repurposing examples available in the first version of repoDB (repoDB.v1, released in March 2017), while the assessment of our predictions (i.e. all the results shown in **Fig. 6**) are only based on the new associations incorporated in the latest version (repoDB.v2, released in June 2020). Considering that repoDB only incorporates repurposing cases with preclinical or clinical evidence (mainly coming from DrugCentral <https://drugcentral.org/> and Clinical Trials <https://clinicaltrials.gov/>), all the predictions that we validated with repoDB.v2 are, by definition, supported by experimental evidence. We also verified that among our top predictions, there were repurposing cases that reached clinical trials (**Fig. R5**, top). Additionally, and following the Reviewer's suggestion, we have incorporated the example of two drugs (Verapamil and Ranolazine) that, even if they are both approved for the treatment of Angina Pectoris (UMLS:C0002963), they both reached clinical trials (NCT02823106 and NCT03162120, respectively) as repurposing drugs of two different diseases (**Fig. R5**, bottom). More specifically, we highlight the treatment of Ischemic stroke (UMLS:C0948008) by Verapamil (predicted as the top 1 repurposing by our model) and the treatment of Atrial fibrillation (UMLS:C0004238) by Ranolazine (predicted as the top 2 repurposing by our model). Interestingly, our model suggests Hyperinsulinemia (UMLS:C0020459) as the top repurposing for Ranolazine. While this relation is not included in repoDB, we have found several studies supporting the relation of Ranolazine with insulin levels (PMID: 23798495, PMID: 26049552, PMID: 28473401). Finally, we would also like to highlight that information about the clinical trials regarding the treatment of Atrial fibrillation by Ranolazine was first posted in May 2017 (<https://clinicaltrials.gov/ct2/show/study/NCT03162120>), two months after the release of the repoDB.v1.

Figure R5. Exploring the repurposing predictions made by the Bioteque. The top panel shows the number of drug-disease repurposing pairs from repoDB.v2 (y-axis) correctly predicted within the top 5 positions (x-axis). Predictions for which repoDB.v2 provides evidence of being in clinical trials are colored in red. The bottom panel shows the predictions made for two drugs, Verapamil (left) and Ranolazine (right). The probability of the predictor is shown on the y-axis while all the 'screened' diseases are ranked on the x-axis. In red we show the prediction validated in repoDB.v2. All the results presented here are based on the Bioteque 'Long' model.

We have now included the discussion of the examples mentioned above, as well as a **supplementary Fig. 6**.

Again for example, the use of drugs with TOP2A as target in Leukemia and Kaposi's Sarcoma is picked out as being a novel relationship between these diseases. However it's not clear what the novelty is because these drugs are widely prescribed for diseases involving rapid cell proliferation which includes both indications.

The Reviewer is right in his/her appreciation that the connection of TOP2A to both Kaposi's sarcoma and Leukemia diseases is known. In fact, it is the existence of this annotation in the databases that allowed the metapath to recapitulate the therapeutic relationship between both diseases. We meant to suggest that, since our metapath recapitulates the therapeutic link between these diseases through TOP2A, it might be this particular protein, or the biological process in which it is involved, what drives the comorbidities observed between these two diseases in orthogonal studies (PMID: 22712016, PMID: 30210797). And, to the best of our knowledge, this TOP2A mediated association of both disorders has never been suggested.

We have now re-written the paragraph to clarify this point, and we thank the Reviewer for pointing out the potentially misleading wording.

Third, given that the novelty of the approach is the embeddings and the ability to use these for future analyses, it would be helpful to have examples of how this could be done. How does one include a new data set in this system? What steps does the user need to take to do this?

We thank the Reviewer for his/her interest in our resource. The Bioteque has been developed as a repository of biomedical descriptors to be used in downstream tasks and, unfortunately, it does not currently contemplate the possibility of creating new embedding spaces based on custom data. Having said that, we agree with the Reviewer that examples of how to deal with the embeddings can increase the impact of our resource and boost the number of potential users. Thus, we now include a jupyter notebook where we illustrate how to (i) generate 2D (interactive) visualizations that can be colored and annotated according to side information (e.g. color cell lines by tissue of origin), (ii) identify similar nodes (close neighbors) for a given entity of interest, (iii) cluster the embedding space, and (iv) build a predictor model trained on our embeddings. We have made this notebook accessible from our GitHub page (<https://gitlab.bnb.irbbarcelona.org/bioteque>) and from the main downloading page of our resource (<https://bioteque.irbbarcelona.org/downloads/demo>). This notebook basically enables most of the downstream analyses presented throughout the manuscript. Future developments of the Bioteque will also include a protocol to create new embeddings but, unfortunately, this cannot be offered as a service, since it will require to have a version of the KG and pretty intensive computational resources.

Minor corrections

- Fig 2C – misspelling of length
- OpenTargets should be Open Targets

We thank the Reviewer for noticing these misspellings and apologies for it. These have been addressed in the updated manuscript.

Reviewer #3 (Remarks to the Author):

Review -- Integrating and formatting biomedical data in the Bioteque, a comprehensive repository of pre-calculated knowledge graph embeddings.

In this manuscript the authors describe a resource, the Bioteque, that provides node embeddings pre-calculated from a comprehensive biomedical graph representing relationships between multiple biological entities. The authors present several examples where these embeddings or vector representations are shown to retain biological information and can be used in machine learning tasks of different purpose. This shows a great effort of data integration built upon previous initiatives such as the Harmonizome or Hetionet and generates a resource that opens multiple possibilities although the usability of the resource could be improved.

We thank the Reviewer for acknowledging our effort, and for recognizing the multiple possibilities the Bioteque may offer for the scientific community.

Embeddings

- The authors generate embeddings using the Node2Vec algorithm and avoid its limitation of not being suitable for heterogeneous networks by representing homogeneous graphs or applying DWPC. Algorithms such as Metapath2Vec (<https://doi.org/10.1145/3097983.3098036>) can handle representation of heterogeneous graphs generating random walks over heterogeneous metapaths and incorporate them into skip-gram model. Could the authors describe why their approach would be preferred?

The Reviewer is right and raises a fair point. In fact, our embedding resource was originally assembled based on metapath2vec embeddings. However, we realized that while metapath2vec is useful to obtain embedding representations for each individual entity appearing in the metapath, this may be inconvenient in some cases. For instance, if we consider the metapath 'Compound-interacts-Gene-phosphorylates-Gene' (CPD-int-GEN-pho-GEN), while the space obtained by our approach will ensure that the observation of a given CPD close to a given GEN means that the CPD is targeting the kinase that phosphorylates the GEN, metapath2vec would not differentiate between the GEN that is directly targeted by the CPD and the GEN that is being phosphorylated by a kinase which is the actual target of the CPD. Notice that kinases can also be substrates of other kinases, hence appearing simultaneously in this metapath as either a direct target of the CPD or as a substrate of a kinase targeted by a CPD. Therefore, using the metapath2vec approach reduces the control of what is actually encoded in the embedding space. Conversely, our approach allows controlling these two cases by differentiating both scenarios into two different metapaths (CPD-int-GEN and CPD-int-GEN-pho-GEN). Furthermore, as DWPC weights each path individually for every source and target node, the final network is already representing those CPD-GEN associations that are more 'enriched' (weighted) in associations, requiring fewer random walker steps to learn the relationship between the source and target nodes. Besides, in metapath2vec the length of the path connecting the source and target nodes directly depends on the length of the metapath, which would require a tailored window size used by the skip-gram model for each metapath. Conversely, as the source and target nodes are always one-hop apart after implementing DWPC, the window size we use captures the same context length regardless of the length of the metapath.

It is important to note that, when the networks are homogeneous or bipartite (as in the case of CPD-int-GEN-pho-GEN), metapath2vec and node2vec are equivalent. In fact, we often use the metapath2vec skip-gram implementation as it proved to be faster than the original node2vec implementation. It is true that metapath2vec offers an alternative skip-gram procedure (called metapath2vec++), where heterogeneous network embeddings are 'normalized' according to the node type. This would be equivalent to node2vec for homogeneous networks but different for bipartite networks. However, we found that this normalization, while it may be convenient in cases where each node type is addressed independently in downstream analysis, also introduces a separation in the embedding space that makes the relations between source and target nodes to be partially lost (**Figure R6**). Since our priority is to describe relationships between the source and target nodes, we opted for not using the metapath2vec++ implementation. Thus, both node2vec and metapath2vec are identical with respect to the networks we have in our resource (homogeneous or bipartite).

We have now added a paragraph to the 'Obtaining embeddings' methods section to clarify our decisions and we explicitly cite the metapath2vec approach.

Figure R6. Assessing the differences between metapath2vec++ and node2vec. Using the same metapath (CPD-int-GEN-ass-DIS), we have obtained 2 embeddings spaces by running either metapath2vec++ (left) or node2vec (right). In the top row, we show the tSNE representation of the embedding space, coloring the nodes according to their entity type. In the bottom row, we show the network preservation score (recapitulation of the original network) using embedding distances. As it is apparent from the plots, the normalization done by metapath2vec++ to each entity type obscures the relation between both entity types, hampering the recapitulation of their associations.

- The authors mention that the aim of the paper is not to benchmark different embedding algorithms and they cite the Node2Vec paper where different shallow embedding methods are compared. However, it would be relevant to compare results from two different learning techniques: shallow vs deep. Algorithms such as Graph Neural Networks can account for more complex network structures than the ones used (heterogeneous, node features) and they may provide more meaningful embeddings and help to represent unseen entities, which may be useful when evaluating a new dataset, compound, etc. These algorithms and their benefits and drawbacks should be at least discussed in the manuscript. I believe the comments in the

discussion regarding “improvement of embedding strategies” points in this direction, but I would suggest making this more explicitly and openly discuss what the limitations of the current approach and how it could be improved.

The Reviewer is again right in his/her appreciation that Graph Neural Networks (GNN) can account for more complex network structures, incorporate node features, and represent unseen entities. Although all these properties are desirable, the superior performance of GNN vastly depends on the availability of meaningful node features and a considerable number of samples (nodes), so that the network information can be properly assimilated. Otherwise, these deep embedding techniques are often surpassed by other embedding techniques such as shallow embeddings (e.g. node2vec) or even classical methods such as matrix factorization algorithms (arXiv:1903.07902). Moreover, GNN methods may perform very differently depending on the network structure (arXiv:2005.10039), requiring a proper fine-tuning of the hyperparameters for each individual case. In our resource, the embedded networks are of different type and size. For instance, our gene co-expression network (GEN-cex-GEN) is homogeneous and comprises 16k nodes each of them having a median of 45 interactions while, on the other hand, our compound-disease treatment network (CPD-trt-DIS) is heterogeneous and barely covers 1k of compounds and 1.5k of diseases having a median of 7 and 4 interactions, respectively. Additionally, the availability of features for some of our entities is not evident (e.g. which feature should be used for a pertubagen (PGN) entity such as an overexpression vector) and, even if they are, they might not be appropriate for every metapath (e.g. note that in Supplementary **Fig. 3** some entities may be ‘identical’ or different depending on the network we are embedding). Therefore, considering that we are embedding more than 1,000 different networks, we decided to provide the first release with a method that has proven to be more consistent across different networks. However, even if GNN are not (at the moment) our preferred method for the systematic embedding of our resource, our descriptors can be recycled as node features for downstream analysis modeled by these GNN structures. For instance, one could use a GNN to predict drug-cell sensitivity interactions using a CPD-CLL network and Bioteque embeddings as features of the nodes (e.g. CPD-int-GEN and CLL-gex-CLL).

We do not discard using GNN or other embedding methods in the future, especially considering the fast growth that the field is having in the last few years (PMID: 34626922). And we completely agree with the Reviewer that, although this is partially mentioned in the discussion section, we do not explicitly refer to GNN methodologies and we do mention the advantages of these methods when successfully implemented. We have extended the discussion to accommodate these interesting points, and thank the Reviewer for pointing them out.

Data integration

- Some of the integrated resources provide diverse scores to weight the associations, i.e STRING, DisGENET, COMPARTMENTS, etc. In the manuscript is not mentioned whether the authors used this information and how. If they did not use these scores, it would be relevant to evaluate how low-confidence scores influence the structure of the graph and thus the embeddings generated.

We have considered the weighted data provided by each resource independently, following recommendations made in the original publications (when available), or based on other studies

that already have explored the proper choice of a particular cutoff. Additionally, when data collections provide different sources of confidence (e.g. DisGeNET and Gene Ontology provide associations from ‘curated’, ‘inferred’, and ‘text-mining’ sources) we split these datasets into subsets (e.g. we have disgenet_curated, disgenet_inferred and disgenet_textmining, each of them including associations from different confidence). In all the cases, we use the curated version as the ‘reference dataset’ for the embedding space. The only exception is DisGeNET, where the reference set comprises the union of ‘disgenet_curated’ and ‘disgenet_inferred’ associations, since we observed that the addition of the ‘inferred’ category did not perturb the characteristics observed in the curated dataset (**Fig. R7**), while enabled us to increase the coverage of diseases (from 5441 to 5810) and genes (from 8431 to 10831) in the embedding space.

Figure R7. Comparing the embedding space of the three DisGeNET versions gathered in the KG. **A**) In the heatmap we show how the embedding space of each dataset (rows) recapitulates (AUROC) each of the datasets (columns). Notice how while the curated version recapitulates, to some extent, the inferred version (AUROC: 0.70), it is not able to recapitulate those associations provided by the textmining category (AUROC: 0.56). **B**) We produced 3 embedding versions, including only the curated associations (red), the curated plus the inferred associations (blue), and the curated plus the inferred plus the textmining associations (green). For each embedding version, we measure how the space recapitulates other DIS-DIS associations (left) and GEN-GEN associations (right). For the DIS-ass-GEN (or GEN-ass-DIS) recapitulation, we use the associations provided in the DisGeNET curated network. Importantly, notice how the addition of inferred associations to the curated version (blue) does not largely perturb the ‘recapitulation profile’ observed when only embedding the curated associations (red). However, the addition of text mining data reduces the recapitulation of other data, being significantly noticeable in the recapitulation of the curated associations from DisGeNET.

Although the specific cutoffs used to treat each individual dataset are provided in the Bioteque web (<https://bioteque.irbbarcelona.org/sources>), we agree with the Reviewer that the manuscript would benefit from having this detailed information explicitly. Therefore, we have included this information as part of **Supplementary Table 3** of the manuscript to clarify how we processed each dataset for its integration, and we thank the Reviewer for pointing out point this missing information in the m/s.

- Looking at the resources ingested, there are in some cases associations integrated from multiple resources. The authors looked at overlap for instance in GEN-assoc-Disease pairs (Figure 1E), but the comparison of up- and down-regulated genes would be very relevant. Did the authors look at the consistency across expression sources? It would be important to evaluate the quality of the integrated datasets to avoid including inconsistencies that could lead to noise influencing the embeddings.

While the Reviewer is right in his/her appreciation that some associations (e.g. GEN-ass-DIS) included in the KG come from different sources, we generally did not merge these datasets when producing an embedding space. Instead, we embedded these datasets separately, so that we can preserve the peculiarities of each dataset (i.e. the provenance information PMID: 30189889) while avoiding including inconsistencies in the embedding space driven by 'batch effects' appearing after merging different datasets. This strategy allowed, for instance, to obtain and compare three different datasets for the same metapath (e.g. GEN-ppi-GEN in Supplementary Fig. 3C). In Fig. 1E, we show the comparison between datasets providing GEN-ass-DIS and GEN-ppi-GEN associations as they turned out to be the most populated associations in our KG (Fig. 1C). In this case, we chose DisGeNET and Open Targets, as reference datasets for the GEN-ass-DIS, and STRING and IntAct as reference for the GEN-ppi-GEN embeddings, as they are comprehensive with respect to the disease and genes included in the resources and their databases are periodically updated. Unfortunately, the comparison of associations between Disease gene regulation datasets is not possible since the KG only contains one data source providing this information (i.e. CREEDs, PMID: 27667448).

However, in line with the request of the Reviewer, we show in the manuscript how embeddings independently obtained from two different cellular gene regulation sources (i.e. the GDSC and CCLE) indeed display a high degree of agreement in their spaces (Fig. 4D).

- In Figure 1F, it seems that the most popular genes, compounds, and pathways point toward cancer, however, only few cancer types show up as popular. This is also surprising given the focus on this disease of some of the resources included (Achilles project, COSMIC, etc.). Could the authors comment on this?

The Reviewer is absolutely right in his/her intuition. The reason why cancer is not noticeable in Fig. 1F is that we used a de-propagated version of the datasets, where the associations are only kept to the most specific (child) terms of the dataset, according to the Disease Ontology. Given the fact that "Cancer (DOID:162)" has 2,026 child terms in the Disease Ontology, after de-propagating the associations the popularity gets diluted among the different cancer child terms, making only a few of them reach the very top positions. However, when we propagate the associations through the ontology, the popularity of cancer becomes evident, as illustrated in (Fig. R8). We thank the Reviewer for noticing the lack of explanation regarding the treatment of the ontology when creating the figure. To avoid misunderstanding, we have now included these details in the legend of the figure and added the word cloud panels using the propagated datasets in Supplementary Fig. 1.

Figure R8. Assessing the popularity of disease nodes when propagating the associations. Left) Popularity distribution of all the diseases (grey) and 26 general cancer terms (purple), obtained by selecting the direct child terms of “cell type cancer” (DOID:0050687) and “organ system cancer” (DOID:0050686) entities. Notice how ‘cancer popularity’ resembles the background distribution when associations are de-propagated, while it becomes evident when associations are propagated. Right) Popularity Word Cloud when using the propagated version of the datasets.

Usability

- The integration effort is one of the most valuable pieces of this work, however, the graph built is not provided to the community. This graph would be very valuable firstly to evaluate the quality of the graph and secondly to use it for further exploration and investigation. The authors could provide the graph in a standard format or tabular edge lists.

We agree with the Reviewer that making the graphs associations available to the community would be a valuable, and probably widely used, asset. However, the added value of the Bioteque is the collection of graph representations (i.e., the embeddings), and we specifically avoided providing the KG. On the one hand, we need to comply with the licenses associated with the data owners, some of which require registration to access the data (e.g., KEGG, DrugBank or COSMIC) while others forbid the re-distribution of the data if no substantial modifications are applied (e.g. data derived from CLUE <https://clue.io/terms#data>), precisely to prevent its re-distribution in multiple meta-sites. Additionally, and not less important, we want to acknowledge the compilation effort of all the resources used in the Bioteque, and thus we do not want to attract users and citations that rightfully should belong to the primary databases.

Nevertheless, we understand the value of providing the KGs used to construct the Bioteque, and thus, we now provide the networks and DWPC weights for metapaths of length $\geq L2$ (i.e. those specifically built in the Bioteque) in their corresponding downloadable file. Additionally, we have updated our GitHub and web resource documentation to guide the user to reproduce the network dataset accommodated in the KG and embedded in metapaths of length L1 from each individual resource.

- The web interface is not sufficiently promoted in the manuscript, and it is an important part of this work. The plots in the web provide the same information as in some of the figures of the

manuscript, however, they provide very little interaction and very little information. For instance, the 2D projection provides very little information about which entities are close to each other or what are the clusters. It would be very useful to be able to identify the dots in the 2D projection by either hovering or searching for specific entities. Further, other annotations such as tissue information, compound class, etc. could be used to colour genes or compounds to show whether clusters represent biological information.

We acknowledge that these added features would increase the functionalities of our web interface. However, the web interface is not intended to be an exploration/analytical tool, but rather an 'interface browser' where users can query the metapaths of interest and download the corresponding embedding space. Having said that, follow-up work on the Bioteque resource will include a set of applications to interact with the resource and run downstream analysis based on the full collection of embeddings. Along the same lines, we also thank the Reviewer for his/her suggestion about making 2D interactive projections, which we think is a great idea that we will consider for future releases and improvements of our resource.

In the meantime, and to guide/help the users in their downstream analyses, we have prepared a jupyter notebook illustrating how to (i) generate interactive 2D projections that can be colored and annotated with custom side information (e.g., color cell lines by tissue information), (ii) identify those entities that are close to each other in the embedding space (i.e. nearest neighbours), (iii) cluster the embedding space, and (iv) build a predictor model trained with our embeddings. Additionally, we improved our web interface by allowing the visualization of different datasets for a given metapath. Moreover, we added a download section (<http://bioteque.irbbarcelona.org/downloads>), where users can directly download the embeddings as well as other metadata (e.g., the list of embeddings in our resource or the covered entity universe). Moreover, we have also included a jupyter notebook showing how to programmatically download all these data. All jupyter notebooks are accessible from the same Downloads page as well as from our GitHub page.

All these new functionalities are now specified in the manuscript.

- I congratulate the authors for providing a link to the repository where the pre-processing scripts can be found! The documentation regarding how these scripts can be used is very limited and it would still make it difficult to replicate the results obtained. I would recommend the authors to provide enough documentation to be able to run and test the scripts provided making use at least of the README files already included to provide examples.

We thank the Reviewer for his/her positive and encouraging comments and, following his/her recommendation, we have updated the GitHub documentation to specify how all these scripts can be run. Moreover, we now link in our web page (<https://bioteque.irbbarcelona.org/sources>) each specific dataset to its corresponding pre-processing folder in the GitHub. Each pre-processing folder includes either a script or a file with instructions to reproduce the data. Besides, we provide a README file detailing the specific choices we followed to accommodate the data (if any) together with a link to the original publication and data source.

Resource update

- The authors mention that the resource and embeddings will be updated yearly or as soon as there are major new versions of datasets. This is excellent, but it would be good if the authors could detail how new versions will be checked for quality (i.e. “how much” embeddings changed, how predictions deviated, etc) and how they will report these metrics. This would be important to ensure the quality of the resource and the validity of previously used embeddings. This is linked to the previous point about the quality of the data integrated, which has an impact on the representations generated.

We thank the Reviewer for acknowledging our effort to update the resource. As in the current version of the Bioteque, to address the quality of the new versions we will: (i) assess how the new embedding space recapitulates the associations of the previous versions (i.e., how much information from the previous release is lost or has been ‘diluted’ in the new one), and (ii) compare how the recapitulation of the orthogonal information changed between dataset versions (i.e., how much different is the information that is being captured in the new dataset compared to the previous one). The current plotting assets, including general stats of the datasets and ROC curves, will be used for this means. As in this first version, all these measures will be provided independently for each metapath embedding within the corresponding downloadable folder.

Reviewers' Comments:

Reviewer #1:

Remarks to the Author:

The authors have put a lot of efforts in revising mine and other reviewers' points. As a results they have further improved their manuscript.

I am overall very satisfied with this revision and I do believe that the publication of this manuscript will represent a good contribution to the field.

Reviewer #2:

Remarks to the Author:

I thank the authors for comprehensively responding to my review. I particularly enjoyed the detailed jupyter notebook example.

Reviewer #3:

Remarks to the Author:

I would like to thank the authors of the manuscript for providing thorough answers to all the comments and suggestions and for including some of these answers into the manuscript. In my view all the concerns were addressed or clarified, and in my view the additional information improves the manuscript and the usefulness of the resource. I will be happy to see it published.

Point-by-point responses to Reviewer's comments

NCOMMS-22-17971 "Integrating and formatting biomedical data in the Bioteque, a comprehensive repository of pre-calculated knowledge graph embeddings" by Fernández-Torras, et al.

We would like to thank the Editor accepting our manuscript for publication and Reviewers for their critical reading of our revision and their positive feedback.

Please find below a point-by-point response to Reviewers' comments.

Reviewer #1 (Remarks to the Author):

The authors have put a lot of efforts in revising mine and other reviewers' points. As a results they have further improved their manuscript. I am overall very satisfied with this revision and I do believe that the publication of this manuscript will represent a good contribution to the field.

We thank the reviewer for acknowledging the effort we put on addressing the reviewers' points and for his/her positive comments on the contribution of our work to the field.

Reviewer #2 (Remarks to the Author):

I thank the authors for comprehensively responding to my review. I particularly enjoyed the detailed jupyter notebook example.

We thank the reviewer for acknowledging that the reply to his/her review was comprehensive. We are glad to find the reviewer enjoyed our detailed jupyter notebook example.

Reviewer #3 (Remarks to the Author):

I would like to thank the authors of the manuscript for providing thorough answers to all the comments and suggestions and for including some of these answers into the manuscript. In my view all the concerns were addressed or clarified, and in my view the additional information improves the manuscript and the usefulness of the resource. I will be happy to see it published.

We thank the reviewer for acknowledging the thoroughness of our answers to all the comments and suggestions and for his/her kind words regarding the publication of our manuscript.